# Parameter Efficient Continual Learning with Dynamic Low-Rank Adaptation

**Prashant Shivaram Bhat**                                       *p.s.bhat@tue.nl*
*Eindhoven University of Technology (TU/e)*

**Shakib Yazdani**                                 *shakib.yazdani@uni-saarland.de*
*Saarland University*

**Elahe Arani**                                                  *e.arani@tue.nl*
*Eindhoven University of Technology (TU/e)*
*Wayve*

**Bahram Zonooz**                                              *b.zonooz@tue.nl*
*Eindhoven University of Technology (TU/e)*

## Abstract

Catastrophic forgetting has remained a critical challenge for deep neural networks in Continual Learning (CL) as it undermines consolidated knowledge when learning new tasks. Parameter efficient fine-tuning CL techniques are gaining traction for their effectiveness in addressing catastrophic forgetting with lightweight training schedule while avoiding degradation of consolidated knowledge in pre-trained models. However, low-rank adapters (LoRA) in these approaches are highly sensitive to rank selection as it can lead to suboptimal resource allocation and performance. To this end, we introduce PEARL, a rehearsal-free CL framework that entails dynamic rank allocation for LoRA components during CL training. Specifically, PEARL leverages reference task weights[2] and adaptively determines the rank of task-specific LoRA components based on the current task's proximity to reference task weights in parameter space. To demonstrate the versatility of PEARL, we evaluate PEARL across three vision architectures (ResNet, Separable Convolutional Network, and Vision Transformer) and a multitude of CL scenarios, and show that PEARL outperforms all considered baselines by a large margin.

## 1   Introduction

Learning forms the foundation for intelligent systems to adapt to dynamic environments in response to external changes. Unlike conventional deep neural networks (DNNs) that are designed for static data distribution, Continual Learning (CL) is characterized by learning from sequential, dynamic data distribution (Wang et al., 2023b). However, CL has remained a significant challenge for DNNs due to the issue of catastrophic forgetting: A phenomenon in which adaptation to new tasks results in significant deterioration of consolidated knowledge (Masana et al., 2022; De Lange et al., 2021; Ditzler et al., 2015; Robins, 1995). A number of approaches such as rehearsal-based (e.g., Buzzega et al. (2020)), weight-regularization based (e.g., Kirkpatrick et al. (2017)) and parameter-isolation based (e.g., Rusu et al. (2016)) have been proposed in the literature to address the problem of catastrophic forgetting in CL. Although CL is primarily intended to ensure the resource efficiency of model updates (Wang et al., 2023b), the reduction in catastrophic forgetting in these approaches is reliant on increases in computational and memory footprint due to model surrogates and memory buffer. Extrapolating forward this reliance in real world scenarios is both economically and environmentally infeasible.

The increased availability of pre-trained models that demonstrate strong generalization capabilities has led researchers to discover early exploratory methods (Douillard et al., 2022; Jeeveswaran et al., 2023; Gao et al., 2024; Yang et al., 2024), facilitating more efficient integration of new knowledge in CL systems. However, continuous fine-tuning on multiple downstream tasks compromises pre-trained model's generalization capabilities (Liu et al., 2024a). Recently, Parameter Efficient Fine-Tuning (PEFT) CL techniques (Liang & Li, 2024; Zhou et al., 2024b) are on the rise as they provide lightweight training schedule while avoiding degradation of consolidated knowledge in pre-trained models. However, it is not clear how one can select the size of their rank (in case of LoRA (Hu et al., 2021)) while their performance is very sensitive to rank selection (Valipour et al., 2022). Although choosing a higher rank usually results in optimal performance, it significantly increases the number of learnable parameters in longer task sequences when using large pre-trained models. Task arithmetic (Ilharco et al., 2023), which enables model compression using simple arithmetic operations with task vectors (difference between pre-trained and fine-tuned weights) (Chitale et al., 2023), allows for selection of higher rank during training without any overhead during inference. Task arithmetic assumes that fine-tuning a pre-trained model pushes the weights away towards a semantically relevant direction in the weight space, thus enabling model compression using simple arithmetic operations with task vectors (Chitale et al., 2023). However, task arithmetic suffers from number of shortcomings including finding optimal scaling coefficients and suboptimal performance in the absence of a memory buffer (Zhou et al., 2024c).

Typically, when using arbitrary weights, a higher rank usually facilitates learning new tasks but tends to increase forgetting with significant increase in number of parameters, while lower-rank mitigates forgetting with limited adaptation (Lu et al., 2024; Biderman et al., 2024). There often exists a balance between plasticity and stability at a moderately small (but not too small) rank, which may enhance the benefits of CL (Lu et al., 2024). To this end, we introduce **PEARL** (Parameter-Efficient Adaptive Ranking for lifelong learning with LoRA), a rehearsal-free, parameter-isolation-based PEFT vision CL framework that mitigates catastrophic forgetting through dynamic low-rank adaptation. Specifically, PEARL leverages the information contained in the reference task weights [1] and dynamically determines the rank of task-specific, layer-specific LoRA components during training based on the current task's proximity to reference task weights in parameter space. With our dynamic threshold criterion, a close proximity would result in a lower rank allocation and maximum information re-use while a distant proximity results in a larger rank selection, thereby finding an optimal balance between learning and forgetting. To demonstrate the versatility of PEARL, we perform extensive analyses across various datasets, multiple CL scenarios and different architectures, an overview of which can be found in Appendix B.1. Our results demonstrate that PEARL variants consistently outperform the evaluated baselines in all scenarios by a huge margin. Our contributions are as follows:

- We propose PEARL, a generic PEFT vision CL framework that mitigates catastrophic forgetting in CL through dynamic low-rank adaptation. PEARL is rehearsal-free and grows modestly with each task (Table 8) by assigning optimal rank for task-specific, layer-specific LoRA components, resulting in maximum information reuse.

- We propose a simple, dynamic threshold criterion (Eqn. 3) guided by the current task's proximity to the reference task weights in the parameter space.

- We conduct extensive experiments across 3 different architectures (ResNet, Separable Convolutional Network, and Vision Transformer), 8 distinct settings, 2 CL scenarios, and 2 CL metrics to clearly demonstrate that PEARL significantly outperforms the considered baselines. To further substantiate our claims, we provide 10+ comprehensive analysis in Appendix C.

## 2 Related Works

**Exemplar and Non-Exemplar Based CL**: Experience Rehearsal (ER), which stores and replays samples from previous tasks, has remained a leading approach to addressing the issue of catastrophic forgetting in CL. Several works build upon ER: DER++ (Buzzega et al., 2020) proposed a consistency regularization technique to enhance stability and generalization. Guided by the GWT, TAMiL (Bhat et al., 2023) and AGILE (Bhat

---

[1]We use *reference task* interchangeably with pre-training when available. Otherwise, it refers to the first task in CL.

et al., 2024) incorporate information reuse and task-specific, parameter-efficient adapters that re-project the backbone features onto a global workspace. However, these approaches face additional computational and memory overhead due to model surrogates and experience rehearsal. Since storing samples from previous tasks is not always feasible, non-exemplar-based approaches are gaining traction. Non-Exemplar Class-IL (NECIL) is particularly important in scenarios where data confidentiality is crucial due to privacy or security concerns, and where data storage has a limited lifespan. Early contributions to this domain include LwF (Li & Hoiem, 2017) and EWC (Kirkpatrick et al., 2017), both of which employ regularization techniques to reduce catastrophic forgetting. More recent methods, such as PASS (Zhu et al., 2021b) and IL2A (Zhu et al., 2021a), have further advanced NECIL by generating prototypes for previous classes without the need to retain the original images. SSRE (Zhu et al., 2022) introduced a reparameterization method to balance old and new knowledge, along with additional self-training that utilizes external data as an alternative to exemplars. FeTrIL (Petit et al., 2023) proposed a framework that combines a fixed feature extractor with a pseudo-feature generator using geometric transformations to achieve a better stability-plasticity balance. FeCAM (Goswami et al., 2024a) addressed the limitations of Euclidean distance when feature distributions are non-stationary and suggested using anisotropic Mahalanobis distance as a substitute for Euclidean distance when employing a frozen feature extractor for prototype classification. Overall, the NECIL benchmark presents a significant challenge as it prohibits the storage of any samples from previous tasks while requiring effective performance in a Class-IL setting. PEARL also aligns with the NECIL benchmark as it operates without any experience rehearsal.

**Parameter Efficient Fine-Tuning (PEFT)** techniques that have emerged in the space of Large Language Models have been quite pivotal in optimizing model performance with little memory and computational overhead. Seminal works such as Adapter modules (Houlsby et al., 2019), Prompt Tuning (Brown, 2020), BitFit (Zaken et al., 2021), LoRA (Hu et al., 2021), and DoRA (Liu et al., 2024b) have demonstrated the efficacy of selective fine-tuning all the while preserving model generalization and enhancing model adaptability. Specifically, LoRA introduces two compact, trainable low rank matrices into each layer's weight matrix, allowing model to adapt to new tasks with minimal memory and computational footprint. Within CL, several approaches leverage PEFT techniques to mitigate catastrophic forgetting: Prompt-based methods such as L2P (Wang et al., 2022e) and DualPrompt (Wang et al., 2022c) introduce task-specific prompts to facilitate learning new tasks while preserving consolidated knowledge. S-Prompt (Wang et al., 2022a) and CODA-Prompt (Smith et al., 2023b) extend these works by employing structural prompts to map discriminative relationships and through Schmidt orthogonalization respectively. In order to maintain an optimal balance between computational complexity and accuracy, adapter-based methods such as SEMA (Wang et al., 2025) and DIA (Li et al., 2025) propose self-expanding, dynamic integration of task-specific adapters. Similarly, EASE (Zhou et al., 2024b) creates task-specific sub-spaces by employing lightweight adapter modules for each new task, thereby maintaining resource efficiency. In contrast to adapter-based methods, InfLoRA (Liang & Li, 2024) inserts a small number of LoRA parameters to re-parameterize the pre-trained weights and shows that fine-tuning these inserted parameters is equivalent to fine-tuning the pre-trained weights within a subspace. C-LoRA (Smith et al., 2023a) incorporates a self-regularization strategy, which penalizes updates to parameters already modified by previous tasks. This mechanism helps preserve prior knowledge and enhances stability across sequential tasks. While these approaches have shown relative success, their implementation is tailored specifically to pre-trained vision transformers and do not consider task dynamics, limiting the potential for information reuse and dynamic resource allocation.

**Task Arithmetic** (Ilharco et al., 2023), centered around task vectors, offers a new paradigm for steering the behavior of pre-trained / reference network towards multiple downstream tasks using task similarity. Inspired by recent works on weight interpolation (Frankle et al., 2020; Ilharco et al., 2022; Ainsworth et al., 2022), task vectors are defined to be the element-wise difference between pre-trained / reference network weights before and after fine-tuning on a particular task. A task vector specifies a direction in the weight space of a pre-trained model, such that movement in that direction improves performance on a given task (Ilharco et al., 2023). As these task vectors can be modified and combined together in simple arithmetic operations, new tasks can be learnt via addition while interfering tasks can be forgotten via negation. Inspired by these findings, Chitale et al. (2023) proposed an approach to continually train transformer-based vision models using low-rank adaptation and task arithmetic. Although task arithmetic is quite lucrative for its simplicity and intuitiveness, it suffers from several shortcomings such as finding optimal scaling coefficients and sub-optimal

performance in the absence of memory buffer to finetune the network after weight aggregation (Zhou et al., 2024c).

To this end, we propose PEARL, a generic PEFT CL framework that leverages task arithmetic to dynamically decide the rank of task-specific, layer-specific LoRA components during training. PEARL is tailored for custom, compact CL, enabling maximum information reuse and modest growth in number of parameters with new tasks in order.

## 3 Method

Our CL setup consists of $t \in \{1, 2, .., K\}$ sequential tasks wherein the CL model $\Phi_\theta$, parameterized by $\theta$, is expected to perform equally well on both current and previous tasks. Each task entails a task-specific independent and identically distributed data distribution $\mathcal{D}_t$ with $\{(x_i, y_i)\}_{i=1}^N$ pairs. In rehearsal-free approaches, access to previous task data distributions is limited when learning a new task. This limitation makes CL especially challenging, as it becomes difficult to maintain performance on older tasks without access to their data distributions. Traditional parameter-isolation methods typically employ task-specific, parameter-intensive sub-networks that are learned in isolation, resulting in a substantial memory and computational footprint. Thus, to strike a balance between model size and performance in rehearsal-free approaches, it is essential to re-use information and enhance the learning of new tasks through parameter-efficient sub-networks. To this end, PEARL is designed to leverage the information aggregated during the reference task training through the dynamic allocation of parameter-efficient low-rank adapters. In the subsequent sub-sections, we detail out the different stages of PEARL development.

### 3.1 Information aggregation during reference task training

Our CL model parameterized as $\theta = \cup_{t=1}^k \theta^{(t)} \cup \theta^r = \cup_{t=1}^K \{f_{\theta^t}, g_{\theta^t}\} \cup \{f_{\theta^r}, g_{\theta^r}\}$, consists of a disjoint set of task-specific parameters and a set of reference task parameters respectively. The feature extractor $f_{\theta^t}$ incorporates low-rank adapters and normalization layers (only in CNNs), while the classifier $g_{\theta^t}$ encompasses all classes pertinent to the respective task. In contrast, the reference task feature extractor $f_{\theta^r}$ typically comprises of a larger network that integrates convolutional or self-attention layers, pooling layers, and normalization layers, with the classifier $g_{\theta^r}$ representing the classes to be learned during the reference task training. To restrict the empirical risk, PEARL minimizes the following objective during reference task training:

$$\mathcal{L}_r = \frac{1}{|b|} \sum_{(x,y)\sim b,\ b\in\mathcal{D}_r} \mathcal{L}_{ce}(\sigma(g_{\theta^r}(f_{\theta^r}(x))), y) \tag{1}$$

where $b$ is a training batch, $\mathcal{L}_{ce}$ is cross-entropy loss, $\mathcal{D}_r$ is the data distribution for the reference task, and $\sigma(.)$ is the softmax function. Since pre-trained models demonstrate strong generalization capabilities, we expect to aggregate generic information that can be re-used across tasks later during CL. In scenarios where pre-training on a large corpus of data is not a possibility, the first task in a CL setup can also be used as a reference task.

### 3.2 Information re-use and dynamic resource allocation during CL

We leverage the knowledge captured in the reference task training and model additional task-specific information pertaining to respective tasks through dynamically allocated low-rank adapters. Let $\mathcal{W}^r \in f_{\theta^r}$ be the weight matrix of a certain layer (either linear, convolutional, or point-wise) belonging to the reference task after its training and $\mathcal{W}^t \in f_{\theta^t}$ be the corresponding weight matrix belonging to current task after finetuning $\mathcal{W}^r$ on $\mathcal{D}_t$. The task vector $\mathcal{W}_c^t$ is given by the element-wise difference between $\mathcal{W}^t$ and $\mathcal{W}^r$ i.e., $\mathcal{W}_c^t = \mathcal{W}^t - \mathcal{W}^r$. In other words, task vector $\mathcal{W}_c^t$ can be thought of as a weight matrix signaling the additional information on top of the information contained in the reference task weights. Therefore, PEARL leverages parameter-efficient low-rank adapters to model task vectors to efficiently learn the current task. To this

---

**Algorithm 1** PEARL (with pre-training)

---

1: **Input**: Pre-trained model $\theta_r = \{f_{\theta^r}, g_{\theta^r}\}$, $\theta = \theta_r$, Data distributions $\{\mathcal{D}_t\}_{t=1}^{K}$, Epochs $E_1$, $E_2$
2: **for** tasks $t \in \{1, 2, .., K\}$ **do**
3:      $\theta_t \leftarrow \{\text{deepcopy}(f_{\theta^r}), g_{\theta^t}\}$
4:      $f_{\theta^t} \leftarrow \{\}$
5:      Fine-tune $\theta_t$ on $\mathcal{D}_t$ for $E_1$ epochs
6:      **for** target layers $l \in \{1, 2, .., L\}$ **do**
7:          Compute $\mathcal{W}_{cl}^t = \mathcal{W}_l^t - \mathcal{W}_l^r$
8:          Decompose $\mathcal{W}_c^t$ using Eqn. 2
9:          Find dynamic rank $k_l$ (Eqn. 3 & Eqn.4)
10:         Initialize LoRA components $B_l$ and $A_l$ (Eqn. 5)
11:         $f_{\theta^t} = f_{\theta^t} \cup \{B_l, A_l\}$
12:      Re-initialize task-specific weights $\{f_{\theta^t}, g_{\theta^t}\}$
13:      $\theta \leftarrow \theta \cup \{f_{\theta^t}, g_{\theta^t}\}$
14:      Fine-tune $\{f_{\theta^t}, g_{\theta^t}\}$ on $\mathcal{D}_t$ for $E_2$ epochs
15: **return** model $\Phi_\theta$

---

end, we decompose $\mathcal{W}_c^t$ (flattened and reshaped to 2D if necessary) using Singular Value Decomposition (SV-Decomposition) as follows:

$$\mathcal{W}_c^t = \mathbf{U\Sigma V^T} = \sum_{i=1}^{n} \sigma_i \mathbf{u}_i \mathbf{v}_i^T \tag{2}$$

where the columns of $\mathbf{U}$ and $\mathbf{V}$ consist of the left and right singular vectors, respectively, $n$ represents the full rank after SV-Decomposition, and $\mathbf{\Sigma}$ is a diagonal matrix whose diagonal entries are the singular values of $\mathcal{W}_c^t$. For $k \in \{1, 2, ..., n\}$, let $\mathcal{W}_{kc}^t = \sum_{i=1}^{k} \sigma_i \mathbf{u}_i \mathbf{v}_i^T$ be the sum truncated after k terms.

**Theorem 3.1** *For any matrix $\mathcal{W}$ of rank at most $k$, $\|\mathcal{W}_c^t - \mathcal{W}_{kc}^t\| \leq \|\mathcal{W}_c^t - \mathcal{W}\|$*

where $\| \cdot \|$ be a unitarily invariant norm. The Eckart-Young-Mirsky Theorem (3.1) implies that the best low k-rank approximation of a matrix can be achieved by selecting top-k singular values and such an approximation maintains the least error in terms of unilaterally invariant norms including the Frobenius and spectral norms. Our novelty lies in finding the best-$k$ that eliminates the need for extensive hyperparameter tuning of the desired $k$ and provides an optimum trade-off between performance and growth in number of learnable parameters. To select the best-$k$, we propose a dynamic threshold on the cumulative explained variance of singular values as follows:

$$\mathcal{T} = \frac{\sum(W_c^t \circ W_c^t)}{\sum(W^t \circ W^t) + \sum(W^r \circ W^r)} \tag{3}$$

where $\mathcal{T}$ represents a normalized squared error between current task and reference task weights, and $\circ$ represents element-wise multiplication. Through $\mathcal{T}$, we approximate the proximity of the current task with respect to the reference task in the parameter space. $\mathcal{T}$ is not guaranteed to lie in $[0, 1]$ and $\mathcal{T}$ can be as large as 2 in extreme cases. Therefore, whenever $\mathcal{T} > 1$, we simply clip it to 1 in our implementation. Once $\mathcal{T}$ is computed, the dynamic rank for the task-specific, layer-specific LoRA can be found as follows:

$$k = \arg\max_j \left( \frac{\sum_{i=1}^{j} \sigma_i^2}{\sum_{i=1}^{r} \sigma_i^2} \geq \mathcal{T} \right) + 1 \tag{4}$$

Equation 3 and 4 ensure that the k is bounded and select the rank in proportion to the cumulative explained variance of singular values. The +1 term in Equation 4 is to ensure the chosen rank is at least one singular

value, preventing a degenerate zero-rank projection and preserving minimal expressive capacity. Subsequently, the low-rank matrices can be initialized from the SV-Decomposition of $\mathcal{W}_c^t$ (flattened and re-shaped to 2D if necessary) as follows:

$$B = \mathbf{U}[:,:k]\,\boldsymbol{\Sigma}[:k] \quad A = \boldsymbol{\Sigma}[:k]\,\mathbf{V}^{\mathsf{T}}[:k,:] \tag{5}$$

As $A$ and $B$ are initialized from SV-Decomposition matrices, at times, the CL model often struggles to find an optimum in the loss landscape resulting in a sub-optimal performance. Therefore, before any further fine-tuning on the current task, we re-initialize the weights of $A$, $B$ along with task-specific normalization (if any) and classification layers. An ablation study on the same can be found in Section 4.5.

### 3.3 Putting it all together

The PEARL framework requires reference task weights to facilitate information re-use across CL tasks. When pre-training is available (Algorithm 1), the pre-trained weights serve as reference task weights. In the absence of pre-training, the first task in CL is considered the reference task (Algorithm 2). For each subsequent CL task, a copy of the reference task weights is fine-tuned for a specified number of epochs ($E_1$) on the current task data using cross-entropy loss. We compute the corresponding layer-specific task vector as the element-wise difference between the fine-tuned weights and the reference task weights. Depending on the backbone architecture, the task vectors are computed for convolutional, point-wise convolutional, or linear projections for the *key* in self-attention, as applicable. The task vector is then decomposed using SV-Decomposition through Equation 2. Since task vectors entail additional information beyond the reference task weights, we model them using dynamic LoRA components. Equations 3 and 4 together define a dynamic rank specific to the current task and layer under consideration. Given the rank $k$, matrices $B$ and $A$ are initialized as described in Equation 5. At times, this initialization may lead to local minima, resulting in suboptimal performance. Therefore, we re-initialize the LoRA matrices, along with any task-specific normalization and classification layers, before further fine-tuning on the current task using cross-entropy loss for another $E_2$ epochs. This process is repeated for all target layers across the backbone network. After fine-tuning, these task-specific parameters are kept frozen.

Our CL model is trained simultaneously on both Class-IL and Task-IL settings, as their training regimes are identical. During inference in the Task-IL setting, $\theta_t = \{f_{\theta^r}, f_{\theta^t}, g_{\theta^t}\}$ is selected based on the given task-id $t$, and its output is inferred for maximum activation. As task-id is not available in Class-IL inference, each test image passes through every sub-network $\theta_t$, and their respective classifier ($g_{\theta^t}$) outputs are concatenated and then inferred for maximum activation. For more details on training and inference in Class-IL and Task-IL settings, refer Sections D.2, and D.3.

## 4 Results

**Experimental settings:** We evaluate PEARL across two distinct scenarios of CL: Class-IL and Task-IL, a complete overview of which can be found in Appendix B.1. To facilitate this evaluation, we consider a comprehensive array of baseline methods: (i) exemplar-based approaches that involve information reuse and task-specific adapters (e.g., TAMiL, AGILE), (ii) NECIL approaches (e.g., PASS, FeTriL, FeCAM), (iii) parameter-isolation approaches that operate within a fixed capacity (e.g., NISPA) and those that expand beyond it (e.g., PNN, PAE), and (iv) PEFT approaches for CL (e.g., L2P, C-LoRA, InfLoRA, SEMA, ADA). We also provide a justification for the choice of these baseline in Appendix B.2. The first three categories utilize randomly initialized ResNet-18 as their backbone, whereas PEFT approaches leverage the ViT-Base-16 model pre-trained on ImageNet-21k. It is important to note that pre-training is often a luxury for smaller, custom models. Consequently, in accordance with the training protocols of the baseline methods, we employ randomly initialized convolutional models, while the ViT model is pre-trained. To demonstrate the versatility of PEARL across different architectures, we explore three distinct backbones: (i) Convolutional networks such as ResNet-18 (R18) and a ResNet-18-like architecture incorporating Blueprint Separable Convolutions (BSC) (Li et al., 2022b), and (ii) the Vision Transformer (ViT) Vit-Base-16 pre-trained on ImageNet-21k. For each subsequent task, we adapt convolutional layer, point-wise convolutional layer and linear layer projecting *key*

Table 1: Comparison of Class-IL performance of exemplar-based and NECIL approaches across different datasets and number of tasks. Baselines use randomly initialized ResNet-18 as a backbone. Exemplar-based approaches use a buffer size of 200. NECIL results are taken from Goswami et al. (2024b), while for the rest, we report an average of 3 runs. Best results are marked in bold while second best are underlined.

| Method | Venue | Seq-CIFAR100 | | | | Seq-TinyImageNet | | | |
| | | T = 5 | | T = 10 | | T = 5 | | T = 10 | |
| | | $A_T\%$ | $\bar{A}\%$ | $A_T\%$ | $\bar{A}\%$ | $A_T\%$ | $\bar{A}\%$ | $A_T\%$ | $\bar{A}\%$ |
|---|---|---|---|---|---|---|---|---|---|
| Sequential fine-tuning | - | 17.94 | 37.54 | 9.43 | 25.07 | 14.41 | 32.83 | 7.92 | 31.55 |
| ER | - | 21.43 | 41.45 | 14.38 | 35.13 | 14.81 | 33.77 | 8.51 | 23.62 |
| DER++ (Buzzega et al., 2020) | NeurIPS 2020 | 28.8 | 47.89 | 25.42 | 43.94 | 17.75 | 35.89 | 10.15 | 27.26 |
| TAMiL (Bhat et al., 2023) | ICLR 2023 | 41.43 | 58.09 | 32.20 | 49.87 | 26.32 | 43.91 | 20.46 | 40.94 |
| AGILE (Bhat et al., 2024) | CoLLAs 2024 | 45.73 | 57.66 | 33.23 | 42.88 | 20.14 | 43.79 | 20.19 | 43.81 |
| LwF (Li & Hoiem, 2017) | ECCV 2016 | 45.35 | 61.94 | 26.14 | 46.14 | 38.81 | 49.70 | 27.42 | 38.77 |
| IL2A (Zhu et al., 2021a) | NeurIPS 2021 | - | - | 31.70 | 48.40 | - | - | 25.30 | 42.01 |
| PASS (Zhu et al., 2021b) | CVPR 2021 | 49.75 | 63.39 | 37.78 | 52.18 | 36.44 | 48.64 | 26.58 | 38.65 |
| SSRE (Zhu et al., 2022) | CVPR 2022 | 42.39 | 56.57 | 29.44 | 44.38 | 30.13 | 43.20 | 22.48 | 34.93 |
| FeTrIL (Petit et al., 2023) | WACV 2023 | 45.11 | 60.42 | 36.69 | 52.11 | 29.91 | 43.99 | 23.88 | 36.35 |
| FeCAM (Goswami et al., 2024a) | NeurIPS 2024 | 47.28 | 61.37 | 33.82 | 48.58 | 25.62 | 39.85 | 23.21 | 35.32 |
| PEARL (R18) | - | _51.97_ | _63.51_ | _41.11_ | _56.21_ | _40.26_ | _51.27_ | _35.22_ | _47.95_ |
| PEARL (BSC) | - | **54.48** | **65.10** | **45.92** | **59.22** | **41.31** | **52.32** | **36.45** | **48.64** |

in self-attention using dynamic low-rank adaptation in each of these backbones respectively. More information on backbones and their settings can be found in Appendix D.

**Metrics**: In line with the established benchmarking protocol, we assess the model's performance using $A_\tau$, representing the accuracy following the $\tau$-th training stage. Notably, we utilize $A_T$, the performance metric at the conclusion of the final stage, along with $\bar{A} = \frac{1}{T} \sum_{\tau=1}^{T} A_\tau$, which computes the average accuracy across all incremental stages. Additionally, we also provide forgetting metrics for our models in Table 11.

### 4.1 Comparison with convolutional CL approaches

Table 1 presents a comparison of PEARL against exemplar-based and NECIL approaches across four different Class-IL settings. We report results for both of our convolutional variants. Since the backbone is randomly initialized without pre-training, PEARL convolutional variants use the first CL task as the reference task and assign LoRA components for subsequent tasks based on their proximity to the first task in the parameter space (Algorithm 2). As shown, PEARL (R18) outperforms all considered baselines without relying on experience rehearsal while exhibiting modest model growth with an increasing number of tasks (see Table 8). Although a larger buffer size might enhance the performance of rehearsal-based approaches, any improvements would likely be outweighed by the increased computational and memory costs (Vijayan et al., 2023b). Additionally, NECIL approaches do not always re-use information across tasks, which often leads to subpar performance. In contrast, PEARL capitalizes on the information contained in the first task for every subsequent task, achieving an optimal balance between performance and model size. While the absence of a pre-trained model may limit the generalizability of the first task, the framework's ability to reuse information and dynamically allocate resources allows PEARL to efficiently assimilate information from both the first and current tasks, resulting in superior performance across tasks.

Likewise, PEARL (BSC) outperforms all baseline methods, including PEARL (R18) in all Class-IL settings. PEARL (BSC), a convolutional variant that employs separable convolutions, is significantly smaller in size compared to other baselines due to its parameter-efficient design. In spite of its modest size, the separation of point-wise and depth-wise operation enables efficient integration of information in BSCs with little room for redundant / interfering information. Coupled with dynamic rank allocation for LoRA components for point-wise filters, PEARL (BSC) enjoys higher performance across all Class-IL scenarios. The trend remains almost similar in Task-IL scenarios (Refer Table 13) as well with PEARL variants outperforming considered baselines by a huge margin.

Table 2: Class-IL performance comparison on ImageNet-R across 5, 10, and 20 task sequences. Baseline results with standard deviation are taken from Liang & Li (2024) and that of PEARL are averaged over 3 runs with errors indicating standard deviation across these runs. The rest of the results are taken from their respective manuscripts. Best results are highlighted in bold. SeqLoRA is a sequential learning baseline which fine-tunes LoRA components on multiple tasks without any additional support.

| Method | Venue | T = 5 | | T = 10 | | T = 20 | |
|---|---|---|---|---|---|---|---|
| | | $A_T\%$ | $\bar{A}\%$ | $A_T\%$ | $\bar{A}\%$ | $A_T\%$ | $\bar{A}\%$ |
| SeqLoRA | - | $70.96 \pm 0.25$ | $79.14 \pm 0.32$ | $64.32 \pm 0.09$ | $74.78 \pm 0.29$ | $56.98 \pm 0.29$ | $69.29 \pm 0.26$ |
| L2P (Wang et al., 2022e) | CVPR 2022 | $64.13 \pm 0.78$ | $68.66 \pm 0.41$ | $62.54 \pm 0.24$ | $67.98 \pm 0.27$ | $57.92 \pm 0.28$ | $64.57 \pm 0.29$ |
| DualPrompt (Wang et al., 2022d) | ECCV 2022 | $67.88 \pm 0.17$ | $71.16 \pm 0.31$ | $65.41 \pm 0.52$ | $69.39 \pm 0.43$ | $61.00 \pm 0.72$ | $65.80 \pm 0.67$ |
| CODA-P (Smith et al., 2023b) | CVPR 2023 | $73.09 \pm 0.21$ | $76.91 \pm 0.21$ | $71.47 \pm 0.35$ | $75.82 \pm 0.29$ | $67.28 \pm 0.30$ | $72.34 \pm 0.17$ |
| C-LoRA (Smith et al., 2023a) | CoRR 2023 | $75.85 \pm 0.31$ | $78.85 \pm 0.34$ | $71.89 \pm 0.45$ | $75.33 \pm 0.28$ | $65.71 \pm 0.60$ | $70.63 \pm 0.85$ |
| LAE (Gao et al., 2023b) | ICCV 2023 | $73.84 \pm 0.14$ | $77.29 \pm 0.45$ | $71.70 \pm 0.39$ | $76.71 \pm 0.10$ | $66.98 \pm 0.35$ | $73.72 \pm 0.05$ |
| InfLoRA (Liang & Li, 2024) | CVPR 2024 | $77.52 \pm 0.37$ | $82.01 \pm 0.12$ | $75.65 \pm 0.14$ | $80.82 \pm 0.24$ | $71.01 \pm 0.45$ | $77.28 \pm 0.45$ |
| EASE (Zhou et al., 2024b) | CVPR 2024 | 70.80 | - | 73.37 | - | 78.80 | - |
| SEMA (Wang et al., 2025) | CVPR 2025 | 79.78 | 84.75 | 78.00 | 83.56 | 74.53 | 81.75 |
| DIA (Li et al., 2025) | CVPR 2025 | - | - | 79.03 | 85.61 | 76.32 | 83.51 |
| TSVD (Peng et al., 2025) | ICLR 2025 | 73.58 | - | 74.55 | - | 79.13 | - |
| PEARL (ViT) | - | $\mathbf{91.97 \pm 0.37}$ | $\mathbf{93.78 \pm 0.36}$ | $\mathbf{88.76 \pm 0.69}$ | $\mathbf{91.67 \pm 0.69}$ | $\mathbf{81.32 \pm 0.79}$ | $\mathbf{86.75 \pm 0.39}$ |

## 4.2 Comparison with PEFT CL approaches

As PEARL entails dynamic allocation of LoRA components, we compare PEARL with state-of-the-art PEFT CL methods across 5, 10, and 20 tasks in ImageNet-R (Hendrycks et al., 2021), in Table 2. The experiment is conducted using the ViT-B/16 backbone (Dosovitskiy et al., 2021) pre-trained on ImageNet-21k. As can be seen, PEARL (ViT) shows a significant improvement in both incremental and final accuracy, with absolute gains ranging from **9**% to **14**%. The performance improvement is more pronounced in 5 task sequence and slightly decreases thereafter as we move towards 10 and then to 20 task sequences. Among considered baselines, InfoLoRA is the closest work in terms of design and performance, but falls short of PEARL (ViT)'s performance by a huge margin across all scenarios. InfLoRA injects a small number of parameters to re-parameterize the pre-trained weights and shows that finetuning these injected parameters is equivalent to finetuning the pre-trained weights within a subspace. Since the reduction matrix $B$ infLoRA is handcrafted rather than being learnt from the training data, InfLoRA suffers a performance degradation compared to PEARL (ViT). Additional comparisons between our method and adapter-based as well as prompt-based approaches can be found in Table 14 and Table 15.

As LoRA components can be plugged in place of any learnable layer, it is imperative to understand the impact of such a low-rank adaptation for different kinds of target layers in ViT. To this end, Table 3 provides an overview of Class-IL performance on ImageNet-R (10T) for different target modules in PEARL (ViT). Our ablation study suggests that adapting linear layers projecting *key* in self-attention yields the best performance across tasks. Adapting *query* modules stands at second when considered individually and third overall. Adapting *values*, either in isolation or in combination with other target modules, yields subpar performance. Although a combination of *query* and *key* is on par with the best results, it entails higher memory footprint when training on longer task sequences. Therefore, throughout our PEARL (ViT) experiments, we apply task-specific LoRA components only to *key* modules.

## 4.3 Comparison with fixed rank as a baseline

The PEFT CL approaches offer lightweight training schedules while effectively retaining the consolidated knowledge of pre-trained models. However, LoRA components in these methods are particularly sensitive to rank selection, which can lead to suboptimal resource allocation and diminished performance (Valipour et al., 2022). Table 4 summarizes the average model size and performance metrics across various ranks for LoRA components. As the rank increases, performance improves at the expense of increased model size; however, beyond a certain rank, the performance gains do not really justify the additional increase in model size. Consequently, tuning the rank is essential to strike the right balance in this trade-off where model growth

Table 3: Class-IL performance comparison of low rank adaptation of different target modules in ViT on ImageNet-R (10T). The best results are highlighted in bold while the second best are underlined.

| Target LoRA Modules | $A_T$ (%) | $\bar{A}$ (%) |
|---|---|---|
| Query | $87.43 \pm 0.41$ | $90.85 \pm 0.17$ |
| Key | $\mathbf{88.76 \pm 0.69}$ | $\underline{91.67 \pm 0.69}$ |
| Value | $11.32 \pm 2.21$ | $20.50 \pm 3.15$ |
| Query & Value | $23.96 \pm 0.80$ | $38.09 \pm 2.74$ |
| Key & Value | $32.38 \pm 0.51$ | $47.13 \pm 4.17$ |
| Query & Key | $\underline{88.74 \pm 0.50}$ | $\mathbf{91.71 \pm 0.42}$ |

remains modest with optimum performance. To this end, PEARL employs a task-specific, layer-specific dynamic rank allocation enabling an optimal trade-off between model size and performance across different datasets and architectures.

Table 4: Class-IL performance comparison of PEARL with static rank allocation for LoRA components. Bold and underline highlight the nearest upper and lower values compared to PEARL. #Params indicate no.of parameters in millions. As can be seen, PEARL achieves a best trade-off between model size and performance without extensive tuning of rank.

| Rank | R18 | | | BSC | | | ViT | | |
|---|---|---|---|---|---|---|---|---|---|
| | Seq-CIFAR100 (5T) | | | Seq-TinyImageNet (10T) | | | ImageNet-R (10T) | | |
| | #Params (M) | $A_T$ (%) | $\bar{A}$ (%) | #Params (M) | $A_T$ (%) | $\bar{A}$ (%) | #Params (M) | $A_T$ (%) | $\bar{A}$ (%) |
| 2 | 11.44 | $45.56 \pm 0.53$ | $58.61 \pm 0.78$ | 1.86 | $32.54 \pm 0.59$ | $45.33 \pm 0.94$ | 91.17 | $84.80 \pm 0.39$ | $89.09 \pm 0.12$ |
| 4 | 11.63 | $49.46 \pm 0.43$ | $61.45 \pm 0.62$ | 1.98 | $33.50 \pm 0.64$ | $46.3 \pm 0.72$ | 91.54 | $86.18 \pm 0.64$ | $90.05 \pm 0.41$ |
| 8 | $\underline{12.01}$ | $\underline{51.30 \pm 1.22}$ | $62.58 \pm 1.04$ | $\underline{2.23}$ | $\underline{35.10 \pm 0.70}$ | $\underline{47.79 \pm 0.66}$ | $\underline{92.28}$ | $87.52 \pm 0.53$ | $90.94 \pm 0.15$ |
| 16 | $\mathbf{12.74}$ | $\mathbf{52.18 \pm 0.36}$ | $\underline{63.34 \pm 0.62}$ | $\mathbf{2.73}$ | $\mathbf{36.50 \pm 0.15}$ | $\mathbf{49.05 \pm 0.37}$ | $\mathbf{93.75}$ | $\underline{88.27 \pm 0.50}$ | $\underline{91.45 \pm 0.09}$ |
| 32 | 14.21 | $52.96 \pm 0.64$ | $\mathbf{63.75 \pm 0.91}$ | 3.72 | $37.26 \pm 0.29$ | $49.74 \pm 0.66$ | 96.71 | $\mathbf{89.17 \pm 0.61}$ | $\mathbf{92.04 \pm 0.21}$ |
| Dynamic (Ours) | 12.56 | $51.97 \pm 0.93$ | $63.51 \pm 0.75$ | 2.30 | $36.45 \pm 0.54$ | $48.64 \pm 0.43$ | 93.27 | $88.76 \pm 0.69$ | $91.67 \pm 0.69$ |

## 4.4 Comparison with convolutional parameter isolation approaches

Figure 1 illustrates a Task-IL comparison between convolutional PEARL variants and various parameter isolation techniques (PNNs, CPG, PAE) as well as dynamic sparse methods (CLIP, NISPA, PackNet) on the Seq-CIFAR100 (20T) dataset. The final Task-IL accuracies after completing the training on all tasks are presented. Parameter isolation methods tend to exceed the model's capacity, whereas dynamic sparse architectures develop task-specific masks while maintaining a fixed model size. PEARL effectively balances these approaches by employing dynamic resource allocation; it expands beyond the model's capacity, albeit more conservatively than other parameter isolation techniques. The results indicate that both convolutional variants of PEARL deliver superior performance across tasks. We attribute this success to effective information reuse and the appropriate allocation of new resources through dynamic resource allocation.

## 4.5 With and without re-initialization

The PEARL framework entails creation of task-specific, layer-specific LoRA components whose weights are initialized from SV-decomposition of task vectors. As task vectors represent the additional information on top of information contained in the reference task, one would expect LoRA components to benefit from such an initialization. Table 5 compares the effects of weight re-initialization of LoRA components, BN layers (if any), and classification layers following SV-decomposition in PEARL. In the *Without re-init* scenario, the weights of the LoRA components are initialized directly from the SV-decomposition of task vectors and are then fine-tuned for the current task. Conversely, in PEARL, the weights of the LoRA components, as well as the batch normalization (BN) and classification layers, are re-initialized after the creation of the LoRA components.The results indicate that weight re-initialization leads to marginally better performance for ResNet-18 on Seq-CIFAR100 (5T), statistically significant performance improvements for ViT on ImageNet-R (10T), and a remarkable two-fold increase in performance for BSC on Seq-TinyImageNet (10T). Overall, the

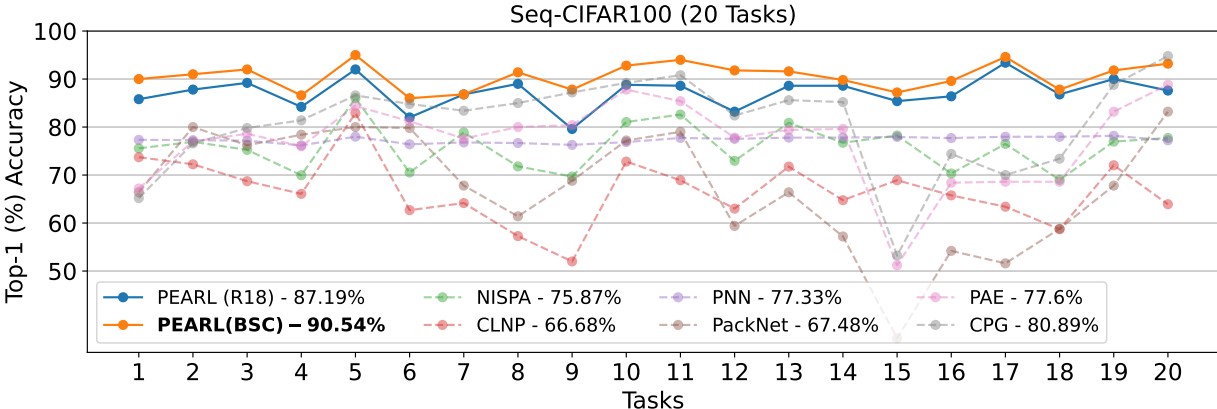

Figure 1: Task-IL performance comparison with parameter isolation approaches on Seq-CIFAR100 (20T). We report the final Task-IL accuracies of each task after training on all tasks. Methods compared are NISPA (Gurbuz & Dovrolis, 2022), PNN (Rusu et al., 2016), PAE (Hung et al., 2019b), CPG (Hung et al., 2019a), PackNet (Mallya & Lazebnik, 2018), and CLNP (Golkar et al., 2019)

evidence suggests that weight re-initialization positively influences PEARL across various CL architectures and settings. We speculate that fresh start due to weight re-initialization helps in avoiding local minima and thus better generalization across tasks.

Table 5: Class-IL performance comparison of PEARL with and without weight re-initialization after creation of LoRA components.

| Dataset | Variant | Metric | Without re-init | With re-init (Ours) |
|---|---|---|---|---|
| Seq-CIFAR100 (5T) | R18 | $A_T$ (%) | $51.34 \pm 0.71$ | $\mathbf{51.97 \pm 0.93}$ |
| | | $\bar{A}$ (%) | $62.37 \pm 1.20$ | $\mathbf{63.51 \pm 0.75}$ |
| Seq-TinyImageNet (10T) | BSC | $A_T$ (%) | $18.71 \pm 0.79$ | $\mathbf{36.45 \pm 0.54}$ |
| | | $\bar{A}$ (%) | $29.25 \pm 0.77$ | $\mathbf{48.64 \pm 0.43}$ |
| ImageNet-R (10T) | ViT | $A_T$ (%) | $78.71 \pm 0.38$ | $\mathbf{83.51 \pm 0.47}$ |
| | | $\bar{A}$ (%) | $84.79 \pm 0.63$ | $\mathbf{88.30 \pm 0.48}$ |

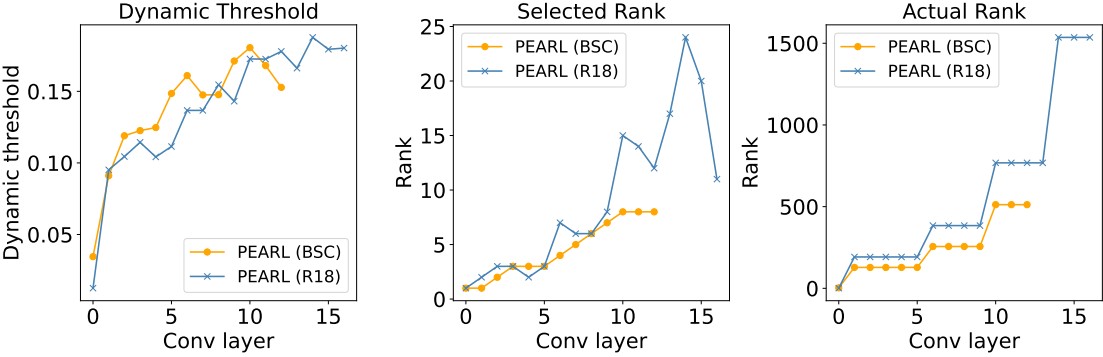

Figure 2: Visualization of final task dynamic resource allocation in Seq-CIFAR100 (5T). (left) Dynamic threshold of different convolutional layers based on the similarity of current task with respect to reference task for deciding the rank. (middle) Selected rank based on dynamic threshold. (right) Actual rank of re-shaped 2-dimensional weight matrix. As can be seen both dynamic threshold and selected rank show more emphasis for later layers, in line with the actual rank.

### 4.6 Dynamic rank allocation across layers

PEARL leverages reference task weights and adaptively determines the rank of task-specific, layer-specific LoRA components based on the proximity of the current task to the reference task weights in the parameter space. Figure 2 illustrates the dynamic rank allocation across layers for convolutional variants for the final task in Seq-CIFAR100 (5T) and that of ViT for the final task in Imagenet-R (10T) can be found in Figure 4. From left to right, Figure 2 presents a detailed overview of the layer-wise dynamic thresholds established based on task proximity, along with the corresponding selected rank and actual rank for our convolutional variants. Consistent with the actual rank, our framework allocates lower ranks to earlier layers and higher ranks to later layers. The dynamic rank allocation is also consistent with established conventions in CNN literature, which suggest that earlier layers capture more generic features, while later layers focus on task-specific information. As a result, earlier layers require less task-specific adaptation, leading to lower rank allocations. Conversely, as we approach the later layers, more resources are necessary to adapt the reference task information to the current task, thereby justifying the higher rank allocation in these layers.

### 4.7 Effect of pre-training on PEARL

To demonstrate the versatility of the PEARL framework, we assessed its performance on two architectures: Convolutional Neural Networks (CNNs) and Vision Transformers (ViTs). Given that ResNet-18 is a widely used model in convolutional CL literature, we employed a randomly initialized ResNet-18 to ensure a fair comparison with baseline models. However, the performance of PEARL (R18) is often limited by the generalization ability of the features captured in the first task. Table 6 provides a comparison of PEARL (R18) with (Algorithm 1) and without (Algorithm 2) pre-training. Thanks to the versatility of PEARL, the framework can work with or without pre-training. As can be seen, pre-training on Imagenet-1K results in superior performance across tasks and across datasets. Wherever possible, we recommend using a pre-trained backbone while employing PEARL for better generalization across tasks.

Table 6: Class-IL performance comparison of PEARL (R18) with and without pre-training. In the former case, PEARL (R18) is pre-trained on Imagenet-1K. Each result is an average of 3 runs with standard deviation alongside. The best results are highlighted in bold.

| Method | Seq-CIFAR100 (5T) | | Seq-TinyImageNet (10T) | |
|---|---|---|---|---|
| | $A_T\%$ | $\bar{A}\%$ | $A_T\%$ | $\bar{A}\%$ |
| Without pre-training | $51.97 \pm 0.93$ | $63.51 \pm 0.62$ | $35.22 \pm 0.27$ | $47.95 \pm 0.34$ |
| With pre-training | $\mathbf{55.34} \pm 0.25$ | $\mathbf{65.97} \pm 0.44$ | $\mathbf{46.15} \pm 0.54$ | $\mathbf{58.50} \pm 0.71$ |

### 4.8 Effect of random class ordering

The Table 7 displays the Class-IL performance on ImageNet-R (10T) under five different class orders, which differ from those presented in Table 2. In this setting, we randomize class assignments across tasks by setting different random seeds, allowing us to assess the impact of class ordering on PEARL. Although the final Class-IL accuracy ($A_T\%$) is slightly lower than the previously reported result of $88.76 \pm 0.69$, the results remain consistent and robust across these new class orders, highlighting the method's stability. Furthermore, both incremental and final accuracies show stability across varying class orderings, reinforcing the reliability of the approach. Even across all different class orderings, PEARL (ViT) consistently outperforms the considered baselines in Table 2 by a huge margin. In conclusion, the findings indicate that PEARL maintains robust performance and generalizability across varying class orders, affirming its reliability as an effective framework for CL, even in randomized conditions.

### 4.9 Growth in model size

Table 8 presents a comparison of parameter growth between PEARL and the complete parameter isolation method. As mentioned earlier, complete parameter isolation involves 100% resource allocation, meaning

Table 7: Class-IL performance comparison on ImageNet-R (10T) across different class orders. Each class order indicates a random mix of classes across tasks. As can be seen, random class ordering has minimal impact on the functioning of PEARL (ViT).

| Class Order | # Params (Millions) | $A_T\%$ | $\bar{A}\%$ |
|:---:|:---:|:---:|:---:|
| 1 | 92.78 | 86.63 | 91.24 |
| 2 | 93.46 | 86.76 | 92.24 |
| 3 | 93.41 | 86.50 | 91.54 |
| 4 | 93.26 | 86.73 | 91.69 |
| 5 | 92.83 | 86.86 | 91.77 |
| **Average** | **93.14** | **86.70** | **91.70** |

each task utilizes a separate backbone network. We regard the complete parameter isolation method as an upper bound in terms of both performance and model growth. This comparison is especially relevant because PEARL is derived from complete parameter isolation principles. However, it distinguishes itself through efficient information reuse and dynamic resource allocation, allowing for effective management of model growth. The results show that PEARL requires significantly fewer parameters than the complete parameter isolation method, highlighting its efficiency. Moreover, PEARL demonstrates scalability, maintaining strong performance even in the context of longer task sequences. This positions it as a more resource-efficient solution for continual learning tasks compared to traditional parameter isolation approaches.

Table 8: Comparison of parameter growth (in Millions) in PEARL with respect to complete parameter isolation in different number of tasks.

| Dataset | Backbone | Method | 5 Tasks | 10 Tasks | 20 Tasks |
|:---:|:---:|:---:|:---:|:---:|:---:|
| Seq-CIFAR100 | R18 | Complete Param. Isolation | 55.89 | 111.73 | 223.42 |
| | | PEARL | **12.56** | **13.27** | **13.55** |
| | BSC | Complete Param. Isolation | 6.32 | 12.60 | 25.14 |
| | | PEARL | **1.63** | **1.89** | **2.30** |
| ImageNet-R | ViT | Complete Param. Isolation | 452.86 | 905.72 | 1,811.44 |
| | | PEARL | **92.46** | **93.27** | **95.35** |

## 5 Conclusion

We proposed PEARL, a rehearsal-free, parameter-isolation-based PEFT CL framework that mitigates catastrophic forgetting through dynamic resource allocation. Specifically, PEARL leverages reference task weights and adaptively determines the rank of task-specific, layer-specific LoRA components based on the current task's proximity to reference task weights in parameter space. With our dynamic threshold criterion, a close proximity would result in a lower rank allocation and maximum information re-use while a distant proximity results in a larger rank selection. To demonstrate the versatility of PEARL, we performed extensive evaluation with three different architectures on a multitude of CL scenarios and showed that these variants outperform all considered baselines by a significant margin. Extending our work to more realistic settings such as general CL scenarios where task boundaries are not known at training time, and exploring alternatives for efficient estimation of task vectors without the need to fine-tune the entire pre-trained model are some of the useful research directions for this work.

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

## A  Limitations and Future work

We provided extensive analysis to exhibit the effectiveness of PEARL across multitude of datasets, settings and scenarios in CL. However, PEARL has several limitations: Firstly, as a parameter isolation approach, PEARL scales with the increasing number of tasks. Although this growth is limited, it still results in additional parameters for each new task, leading to a larger model. Moreover, with every new task, each image has to pass through every sub-network for Class-IL inference, resulting slower inference. Secondly, PEARL assumes knowledge of task boundaries, which are not always available in real-world scenarios. This assumption is common among parameter isolation approaches, as new parameters are typically introduced at the onset of each task. Thirdly, PEARL requires fine-tuning a full pre-trained model on the current task for several epochs. As model sizes grow, this may become impractical on resource-constrained devices. Finally, task vector initialization in PEARL assumes a precise measure of task proximity, but in practice, there can be multiple proximity criteria can be explored and validated. In our future work, we aim to explore alternatives for defining task boundaries and efficient estimation of task vectors without the need to fine-tune the entire pre-trained model.

## B  Additional Information

### B.1  Overview of experiments

We provide a comprehensive overview of the experiments conducted to evaluate the performance of PEARL across various CL benchmarks. To demonstrate the versatility of the PEARL framework, we assess its performance on two architectures: Convolutional Neural Networks (CNNs) and Vision Transformers (ViTs). Given that ResNet-18 is a widely used model in convolutional CL literature, we employ a randomly initialized ResNet-18 to ensure a fair comparison with baseline models. Furthermore, we extend our experiments to include a ResNet-18-like architecture with blueprint separable convolutions (BSCs). The decision to include a BSC-based network is driven by two key considerations: (i) As CL evolves, there is a growing demand for efficient models that maintain a modest parameter size. BSCs significantly reduce the parameter footprint compared to traditional convolutions, resulting in more compact models; and (ii) By separating the traditional convolution operation into point-wise and depth-wise operations, BSCs allow us to illustrate the generalizability of our approach beyond conventional model architectures. Consistent with baseline methods, these convolutional models are not pre-trained and are initialized randomly. Since pre-training can be a luxury for small custom models, these experiments highlight that pre-training is not necessary for the effective application of PEARL. In these settings, we treat the first task as a reference task and proceed with parameter-efficient fine-tuning starting from the second task. We evaluate PEARL against three types of baselines: exemplar-based methods, NECIL approaches, and parameter isolation techniques. We present results on standard CL benchmarks, including Seq-CIFAR100 and Seq-TinyImageNet, in both Class-IL and Task-IL settings.

Similar to PEARL, parameter-efficient fine-tuning (PEFT) approaches in CL leverage techniques to adapt pre-trained ViT models for new tasks. Consequently, we consider PEFT CL approaches as primary baselines for our PEARL (ViT) version. In alignment with these baselines, we utilize the ViT-base-16 model, which has been pre-trained on the ImageNet-21k dataset. The larger size of ViT-base-16 enables us to experiment with a more extensive dataset. Accordingly, we evaluate PEARL (ViT) on the rendered ImageNet dataset (ImageNet-R) in a Class-IL setting across various task sequences.

### B.2  Justification for considered baselines

To facilitate extensive evaluation of PEARL, we consider a comprehensive array of baseline methods: (i) exemplar-based approaches that involve information reuse and task-specific adapters (e.g., TAMiL, AGILE);

Table 9: Summary of experiments.

| Architecture | Model | Reference task | Setting | CL scenario | Baseline | Metric |
|---|---|---|---|---|---|---|
| Convolutional | - R18
- BSC | - Not pre-trained | - Seq-CIFAR100 (5T, 10T, 20T)
- Seq-TinyImageNet (5T, 10T) | - Class-IL
- Task-IL | - Exemplars based
- NECIL
- Param isolation based | - $\bar{A}$ (%)
- $A_T$ (%) |
| Transformers | - ViT | - Pre-trained | - ImageNet-R (5T, 10T, 20T) | - Class-IL | - PEFT CL | |

(ii) NECIL approaches (e.g., PASS, FeTriL, FeCAM); (iii) parameter-isolation approaches that operate within a fixed capacity (e.g., NISPA) and those that expand beyond it (e.g., PNN, PAE); and (iv) PEFT approaches for continual learning (e.g., L2P, C-LoRA, InfLoRA). Firstly, inspired by the GWT, exemplar-based baselines entail a shared network for information reuse and task-specific attention modules that re-project the representation from the common representation space to the global workspace. Likewise, the PEARL framework includes a shared reference task network for information reuse and task-specific, layer-specific low-rank adapters for orienting towards the current task. Secondly, PEARL is a rehearsal-free framework; it does not store previous task samples either directly or indirectly through generative networks. Therefore, it aligns well with the NECIL benchmark and associated approaches. Thirdly, we note that low-rank adaptation introduces task-specific parameters that are trained on the corresponding task and kept frozen afterwards, akin to parameter-isolation approaches. As PEARL exhibits partial parameter isolation with a shared reference task network and low-rank adapters, we also provide a comparison with parameter isolation approaches. Finally, since PEARL readily utilizes LoRA (Hu et al., 2021), a prominent PEFT technique, we offer an in-depth comparison with PEFT continual learning approaches. In addition, we also provide a discussion on related dynamic low-rank adaptation techniques beyond continual learning in Section B.3.

To the best of our knowledge, PEARL is the first dynamic low-rank adaptation framework tailored specifically for generic or custom-built small-scale models in continual learning. With a variety of the aforementioned baselines and analysis, we provide a comprehensive understanding of PEARL across architectures, datasets, and continual learning scenarios.

### B.3 Discussion on PEFT LoRA counterparts

**PEFT CL approaches with static rank allocation**: Among PEFT continual learning approaches, InfLoRA (Liang & Li, 2024) and C-LoRA (Smith et al., 2023a) are prominent techniques that involve low-rank adaptation of pre-trained networks, similar to PEARL. Each of these methods are tailored for CL scenarios and addresses unique challenges. InfLoRA aims to minimize the interference of new tasks on previously learned ones, thereby improving knowledge retention. However, its reliance on a handcrafted reduction matrix $B$, rather than a data-driven learned representation, limits its flexibility. C-LoRA, on the other hand, depends on a pre-trained model as its foundation, making it reliant on the availability of such models for its application. It applies LoRA to both the key and query attention mechanisms, ensuring efficient adaptation for new tasks. Additionally, C-LoRA incorporates a self-regularization strategy, which penalizes updates to parameters already modified by previous tasks. This mechanism helps preserve prior knowledge and enhances stability across sequential tasks. Despite its relative success, caution is warranted when training C-LoRA on large task sequences, as this approach suffers from scalability issues. In line with standard practices in continual learning benchmarks, we provide results for SeqLoRA, a sequential finetuning baseline without any safeguards against catastrophic forgetting. SeqLoRA experiences significant forgetting, underscoring the difficulty of maintaining performance across tasks in continual learning.

The PEARL framework proposed in this paper entails dynamic rank allocation for LoRA components, making it efficient and scalable even with longer task sequences. Secondly, due to efficient dynamic resource allocation, all LoRA parameters are learnable through gradient descent. Unlike InfLoRA, this flexibility allows PEARL to capture the nuances of the corresponding task more effectively without any hindrance. With regard to C-LORA, PEARL does not entail self-regularization to bring forth stability and consequent reduced task interference. We speculate that task dynamics as played out in the parameter space allow PEARL to counteract any task interference, thereby effectively mitigating forgetting across CL tasks. Overall, we find

that dynamic resource allocation coupled with the flexibility to learn task nuances based on task proximity has endowed PEARL with superior performance across a multitude of Cl scenarios.

**PEFT approaches with dynamic rank allocation**: Parameter efficient fine-tuning techniques such as LoRA distribute trainable parameters equally across all layers, which may reduce efficiency when layers vary in their contribution towards task adaptation. To address this, adaptive low-rank adaptation strategies have emerged that dynamically adjust parameter budgets to improve fine-tuning efficiency, similar to PEARL. AdaLoRA (Zhang et al., 2023) adaptively allocates the parameter budget among weight matrices according to their importance score. The importance score is computed by parameterizing the incremental updates in the form of SV-decomposition and the singular values of unimportant updates are pruned. DyLoRA (Valipour et al., 2023) proposed a dynamic low-rank adaptation in which LoRA blocks are trained for a range of ranks instead of a single rank by sorting the representation learned by the adapter module at different ranks during training. More recent methods, such as DoRA (Mao et al., 2024) and Dynamic Rank-Selective LoRA (Lu et al., 2024), have emerged, with DoRA decomposing high-rank LoRA layers into structured single-rank components, while Dynamic Rank-Selective LoRA adaptively assigns ranks to LoRA modules based on their relevance to the current data in CL in vision language models. Although these approaches address a similar challenge as PEARL, these are not tailored for CL in custom, compact models.

AdaLoRA uses SVD to measure the importance of each matrix component, pruning lower singular values. However, this adaptive adjustment introduces a more complex implementation compared to PEARL, and potential instability if rank adjustments are too frequent. DyLoRA addresses the limitations of static rank selection by training LoRA blocks with a range of ranks, randomly sampling a rank during each training iteration and optimizing the model for that rank. Unlike DyLoRA, PEARL does not require setting a predefined rank range before training, offering greater flexibility during training. DoRA introduces a decomposition strategy that splits high-rank LoRA layers into multiple single-rank components, with dynamic rank allocation and a regularization penalty to ensure stable pruning. However, DoRA does not result in significant reduction in number of parameters compared to high-rank LoRA allocation. Dynamic Rank-Selective LoRA introduces dynamic importance weights, represented by a learnable vector, which assigns importance to each rank during training. These weights, inspired by the diagonal elements in the singular value matrix of SV-Decomposition, are learned directly from the data while promoting sparsity by minimizing the $\ell_1$ norm of the importance weights. However, Dynamic Rank-Selective LoRA directly merges parameter updates into the pre-trained weights without explicitly considering correlations between tasks. Overall, the three former approaches are not specifically tailored for CL and lack results for direct comparisons with PEARL variants. Although Dynamic Rank-Selective LoRA focuses on CL, it is tailored for vision-language models, limiting the possibility of a direct comparison with PEARL. To the best of our knowledge, PEARL is a one of the early approaches tailored specifically for efficient custom, compact CL.

### B.4 Potential sensitivity of PEARL

PEARL is quite sensitive to the stability of task vector and noise contained in the resultant SVD decomposition thereof. In cases where task vectors are poorly conditioned, the dynamic rank estimation can be imprecise resulting in over or under allocation of dynamic rank. Figure 5(b) provides a depiction of resource allocation dynamics for ImageNet-R (10T) using PEARL (ViT) across three distinct runs. We note that PEARL relies solely on the proximity of a task relative to the reference task weights in parameter space. It is important to clarify that task proximity, as defined in PEARL, does not correspond to semantic similarity between tasks. Different random seeds can alter the training trajectory, thereby changing the proximity of tasks relative to the reference task. As a result, the distribution of parameter allocation can vary significantly across tasks in different runs.

### B.5 Relation to Orthogonal-Subspace Methods

Orthogonal-subspace approaches (e.g., Wang et al. (2023c); Chaudhry et al. (2020)) address catastrophic forgetting in CL by enforcing geometric constraints that keep updates for new tasks orthogonal to subspaces associated with previous tasks, typically using SVD-based decompositions or orthogonalized gradients. These methods effectively reduce interference but require explicit construction, storage, and maintenance

of task-specific subspaces, causing computational and memory costs to grow with the number of tasks. Moreover, they operate in the full parameter or gradient space, which can be expensive and less practical for large modern architectures. We compare against this line of work because it shares the core objective of preserving prior knowledge while providing task-specific flexibility, making it a natural conceptual baseline for evaluating PEARL. In contrast to explicit orthogonality constraints, PEARL works entirely within the parameter-efficient LoRA space and introduces dynamic rank allocation based on task similarity, allowing it to adapt capacity without building or maintaining orthogonal subspaces. This results in a more scalable and lightweight mechanism for mitigating interference, while still offering the ability to adapt to diverse tasks in a rehearsal-free continual learning setting.

## C Additional Analysis

### C.1 Comparison with complete parameter isolation

At its core, PEARL functions as a parameter-isolation CL framework that dynamically allocates resources based on the current task's proximity to the reference task in the parameter space. The theoretical upper bound for PEARL represents a model with 100% resource allocation for each task. To illustrate this, we compare PEARL with the *Complete Parameter Isolation* method, which entails a separate backbone for each task. For instance, in Seq-CIFAR100 (5T), the Complete Parameter Isolation method would necessitate a distinct ResNet-18 backbone for each task, enabling maximum resource allocation and complete parameter isolation. Table 10 provides comparison of PEARL against complete parameter isolation with maximum resource allocation. Across settings and architectures, PEARL stays close to the performance of complete parameter isolation method while being extremely parameter efficient.

Table 10: Class-IL ($A_T\%$) performance evaluation of PEARL with respect to a model with complete parameter isolation. We treat such as model as an upper bound for PEARL. Efficiency is computed as a ratio of Class-IL ($A_T\%$) Vs #Params (M). As can be seen, dynamic rank allocation leaves minimal footprint while staying close to the performance of complete parameter isolation method.

| Dataset | Backbone | Model | #Params (M) | Class-IL ($A_T\%$) | Task-IL ($A_T\%$) | Efficiency |
|---|---|---|---|---|---|---|
| Seq-CIFAR100 (5T) | R18 | Complete param isolation | 55.89 | **53.57** $\pm$ 0.66 | **80.69** $\pm$ 0.54 | 0.96 |
| | | PEARL | **12.56** | 51.97 $\pm$ 0.93 | 80.04 $\pm$0.54 | **4.14** |
| | BSC | Complete param isolation | 6.32 | **59.22** $\pm$ 0.29 | **83.82** $\pm$ 0.19 | 9.37 |
| | | PEARL | **1.63** | 54.48 $\pm$ 0.35 | 81.55 $\pm$ 0.71 | **33.42** |
| Seq-TinyImageNet (10T) | R18 | Complete param isolation | 111.79 | **38.5** $\pm$ 0.32 | **73.54** $\pm$ 0.53 | 0.35 |
| | | PEARL | **12.32** | 35.22 $\pm$ 0.27 | 71.49 $\pm$ 0.22 | **2.86** |
| | BSC | Complete param isolation | 12.65 | **40.63** $\pm$ 0.48 | **73.3** $\pm$ 0.27 | 3.21 |
| | | PEARL | **2.3** | 36.45 $\pm$ 0.53 | 70.95 $\pm$ 0.44 | **15.84** |
| ImageNet-R (10T) | ViT | Complete param isolation | 905.72 | 87.58 $\pm$ 0.49 | **97.28** $\pm$ **0.19** | 0.09 |
| | | PEARL | **93.27** | **88.76** $\pm$ **0.69** | 97.19 $\pm$ 0.18 | **0.95** |

### C.2 Forgetting analysis

We provide additional metrics in our experiments to comprehensively evaluate the performance of PEARL. Table 11 includes metrics such as forgetting, stability, and plasticity, which offer valuable insights into the model's behavior. It is noteworthy that most baseline models do not report these metrics, limiting our ability to compare their performance across these dimensions. Forgetting measures the model's ability to retain knowledge from previous tasks by calculating the average decline in accuracy for a task at the conclusion of CL training relative to its initial accuracy; lower forgetting values indicate superior knowledge retention. Stability (S) reflects the average accuracy on previously learned tasks at the end of training, illustrating the model's performance on earlier tasks. In contrast, plasticity (P) evaluates the model's capability to learn new tasks effectively, determined by the average accuracy during the initial training of those tasks. The trade-off between stability and plasticity is quantified by the formula $\frac{2 \times S \times P}{P+S}$, which measures how well the

method balances these two aspects. Together, these metrics offer a holistic view of PEARL's performance, emphasizing its strengths and trade-offs in various Class-IL scenarios. Across both Seq-CIFAR100 (5T) and Seq-TinyImageNet (10T), PEARL (BSC) demonstrates superior performance and a more favorable trade-off between stability and plasticity compared to PEARL (R18). A notable exception is observed in the forgetting metric for Seq-CIFAR100 (5T), where PEARL (BSC) experiences slightly higher forgetting. However, the enhanced stability and plasticity in PEARL (BSC) effectively compensate for this increase in forgetting. On ImageNet-R (10T), the PEARL (ViT) method demonstrates strong performance across key continual learning metrics. It achieves high average accuracy ($A_T$) of 88.76%, excellent plasticity (91.67%), and minimal forgetting (3.63%), indicating a well-balanced trade-off between stability and adaptability.

Table 11: PEARL Class-IL performance metrics across datasets.

| Dataset | Method | $A_T$ (%) ↑ | Forgetting (%) ↓ | Stability (%) ↑ | Plasticity (%) ↑ | Trade-off ↑ |
|---|---|---|---|---|---|---|
| Seq-CIFAR100 (5T) | PEARL (R18) | $51.97 \pm 0.34$ | $13.34 \pm 0.39$ | $51.19 \pm 0.68$ | $62.64 \pm 0.90$ | $56.34 \pm 0.77$ |
| | PEARL (BSC) | $54.47 \pm 0.27$ | $14.47 \pm 0.34$ | $53.21 \pm 0.41$ | $66.06 \pm 0.54$ | $59.01 \pm 0.32$ |
| Seq-TinyImageNet (10T) | PEARL (R18) | $35.22 \pm 0.49$ | $14.26 \pm 0.56$ | $35.41 \pm 0.25$ | $48.06 \pm 0.32$ | $40.78 \pm 0.14$ |
| | PEARL (BSC) | $36.45 \pm 0.68$ | $13.44 \pm 0.65$ | $36.56 \pm 0.53$ | $48.54 \pm 0.95$ | $41.70 \pm 0.59$ |
| ImageNet-R (10T) | PEARL (ViT) | $88.76 \pm 0.69$ | $3.63 \pm 0.17$ | $88.18 \pm 0.77$ | $91.67 \pm 0.71$ | $89.89 \pm 0.74$ |

## C.3  Hyperparameter Tuning

PEARL framework provides a robust way to find the best trade-off between model size and performance in PEFT CL. As such, our framework is devoid of any specific hyper-parameters. In order to provide a better picture of PEARL's performance landscape, Table 12 presents Class-IL ($A_T$%) performance for various learning rates. The best results for each dataset and model combination are highlighted in bold, providing a clear indication of the optimal settings. For Seq-CIFAR100 (5T), the results show that a learning rate of 0.001 yielded the highest performance for both PEARL (R18) and PEARL (BSC). Notably, the performance of PEARL (R18) is better than its BSC counterpart in all other learning rates except 0.001 and 0.01. Therefore, when generalizing to datasets beyond those considered in this paper, we urge the readers to extensively tune learning rate of PEARL (BSC) for optimum performance. In the Seq-TinyImageNet (10T) scenario, the optimal learning rate for PEARL (R18) was found to be 0.0001, achieving an $A_T$ of 35.22%. For PEARL (BSC), a slightly higher learning rate of 0.001 led to the best performance of 36.45%. PEARL (BSC)'s performance across different learning rates further confirms the susceptibility of PEARL (BSC) to changing learning rates. For the ImageNet-R (10T) dataset, the results demonstrate that the learning rate of 0.03 provided the best performance for PEARL (ViT), resulting in an $A_T$ of 91.38%. As PEARL (ViT) is pre-trained on ImageNet-21k, it is more robust to changes in the learning rate.

Table 12: Comparison of Class-IL performance ($A_T$ (%)) for various learning rates. The best results for each dataset and model combination are highlighted in bold. The error terms indicate the standard deviation across 3 random runs.

| Dataset | Model | Class-IL ($A_T$%) for different learning rates | | | | | |
|---|---|---|---|---|---|---|---|
| | | 0.1 | 0.01 | 0.03 | 0.001 | 0.005 | 0.0001 |
| Seq-CIFAR100 (5T) | PEARL (R18) | $30.95 \pm 6.12$ | $42.69 \pm 1.78$ | $41.62 \pm 0.76$ | $\mathbf{51.97} \pm 0.93$ | $47.25 \pm 0.28$ | $48.89 \pm 0.28$ |
| | PEARL (BSC) | $10.44 \pm 1.59$ | $52.18 \pm 0.46$ | $40.63 \pm 3.02$ | $\mathbf{54.48} \pm 0.36$ | $47.60 \pm 0.53$ | $45.53 \pm 0.48$ |
| Seq-TinyImageNet (10T) | PEARL (R18) | $14.77 \pm 1.41$ | $27.18 \pm 0.75$ | $26.15 \pm 0.98$ | $34.63 \pm 0.67$ | $28.57 \pm 0.99$ | $\mathbf{35.22} \pm 0.27$ |
| | PEARL (BSC) | $3.47 \pm 1.08$ | $33.93 \pm 0.60$ | $18.69 \pm 1.51$ | $36.22 \pm 0.53$ | $\mathbf{36.45} \pm 0.53$ | $29.30 \pm 0.39$ |
| ImageNet-R (10T) | PEARL (ViT) | $91.23 \pm 0.18$ | $91.06 \pm 0.58$ | $\mathbf{91.38} \pm \mathbf{0.50}$ | $83.51 \pm 0.47$ | $88.76 \pm 0.69$ | $83.51 \pm 0.47$ |

## C.4  Comparison against CL approaches in standard benchmarks.

Table 13 presents a comparison of PEARL convolutional models in both Class-IL and Task-IL scenarios for Seq-CIFAR100 (5T) and Seq-TinyImageNet (10T). Rehearsal-based methods, including those that rely

Table 13: A benchmark comparison with CNN-based prior works on Class-IL ($A_T\%$ ) and Task-IL ($A_T\%$ ). The best results are highlighted in bold while the second best are underlined. The error terms indicate a standard deviation across 3 random runs. The rehearsal-based baselines employ a buffer size of 200.

| Method | Venue | Seq-CIFAR100 (5T) | | Seq-TinyImageNet (10T) | |
|---|---|---|---|---|---|
| | | Class-IL ($A_T\%$ ) | Task-IL ($A_T\%$ ) | Class-IL ($A_T\%$ ) | Task-IL ($A_T\%$ ) |
| JOINT | - | 70.56 ±0.28 | 86.19 ±0.43 | 59.99 ±0.19 | 82.04 ±0.10 |
| SGD | - | 17.49 ±0.28 | 40.46 ±0.99 | 7.92 ±0.26 | 18.31 ±0.68 |
| PNNs (Rusu et al., 2016) | NeurIPS 2016 | - | 74.01 ±1.11 | - | **67.84** ±0.29 |
| PackNet (Mallya & Lazebnik, 2018) | CVPR 2018 | - | 72.39 ±0.37 | - | 60.46 ±1.22 |
| NISPA (Gurbuz & Dovrolis, 2022) | NeurIPS 2022 | - | 65.36 ±2.19 | - | 59.56 ±0.32 |
| SparCL-EWC (Wang et al., 2022b) | NeurIPS 2022 | - | 59.53 ±0.25 | - | 59.56 ±0.32 |
| ER | - | 21.40 ±0.22 | 61.36 ±0.35 | 8.57 ±0.04 | 38.17 ±2.00 |
| DER++ (Buzzega et al., 2020) | NeurIPS 2020 | 29.60 ±1.14 | 62.49 ±1.02 | 10.96 ±1.17 | 40.87 ±1.16 |
| ER-ACE (Caccia et al., 2022) | ICLR 2022 | 35.17 ±1.17 | 63.09 ±1.23 | 11.25 ±0.54 | 44.17±1.02 |
| Co$^2$L (Cha et al., 2021) | ICCV 2021 | 31.90 ±0.38 | 55.02 ±0.36 | 13.88 ±0.40 | 42.37 ±0.74 |
| OCDNet (Li et al., 2022a) | IJCAI 2022 | 44.29 ±0.49 | 73.53 ±0.24 | 17.60 ±0.97 | 56.19 ±1.31 |
| TAMiL (Bhat et al., 2023) | ICLR 2022 | 41.43 ±0.75 | 71.39 ±0.17 | 20.46 ±0.40 | 55.44 ±0.52 |
| TriRE (Vijayan et al., 2023a) | NeurIPS 2023 | 43.91 ±0.18 | 71.66 ± 0.44 | 20.14 ±0.19 | 55.95 ±0.78 |
| AGILE (Bhat et al., 2024) | CoLLAs 2024 | 45.73 ±0.15 | 74.37 ±0.34 | 20.19 ±1.65 | 53.47 ±1.60 |
| PEARL (R18) | - | $\underline{51.97} \pm 0.93$ | $\underline{80.05} \pm 0.54$ | $\underline{35.22} \pm 0.27$ | **71.49** ± 0.23 |
| PEARL (BSC) | - | **54.48** ± 0.36 | **81.55** ± 0.72 | **36.45** ± 0.53 | $\underline{70.94} \pm 0.44$ |

exclusively on experience replay (e.g., ER, DER++), exhibit subpar performance in both Class-IL and Task-IL contexts. In contrast, approaches that utilize one or more model surrogates (e.g., Co$^2$L, OCDNet, TAMiL) demonstrate improved performance, which aligns with the increase in overall model size. Parameter isolation methods (e.g., PNNs, PackNet) that require task-identity at inference show a great promise in Task-IL setting. However, these methods struggle in Class-IL scenarios due to challenges related to inter-task class separation. Conversely, both PEARL models consistently outperform all examined baselines across all scenarios. The robust performance across varied Task-IL settings highlights the versatility and effectiveness of the PEARL framework, underscoring its potential as a leading solution in continual learning tasks.

## C.5   Performance in longer task sequences

It is essential for CL models to excel in longer task sequences, as real-world applications often require them to adapt to a variety of tasks in succession. Figure 3 showcases the Class-IL performance ($A_T\%$) of PEARL over extended task sequences of 5, 10, and 20 tasks in Seq-CIFAR100. Parameter isolation approaches, including PEARL, provide maximum stability by fixing parameters associated with previous tasks. Although there is no capacity saturation, as the number of tasks increases, forgetting becomes more pronounced due to the difficulty of distinguishing between a larger set of classes, leading to a decrease in performance. This is reflected in the steady drop in performance as we progress from 5 to 10 and then to 20 tasks, highlighting the challenges of maintaining knowledge retention in increasingly complex Class-IL task sequences.

## C.6   Dynamic rank allocation across layers in ViT

PEARL introduces a PEFT CL framework that incorporates dynamic rank selection for LoRA components during training. PEARL leverages reference task weights and adaptively determines the rank of task-specific, layer-specific LoRA components based on the proximity of the current task to the reference task weights in the parameter space. Figure 2 illustrates the dynamic rank allocation across layers for convolutional variants for the final task in Seq-CIFAR100 (5T). From left to right, Figure 2 presents a detailed overview of the layer-wise dynamic thresholds established based on task proximity, along with the corresponding selected rank and actual rank for our convolutional variants. Similarly, Figure 4 presents the dynamic rank allocation across layers for the final task in ImageNet-R (5T) for PEARL (ViT). In the ViT model, earlier layers are primarily responsible for extracting low-level image features. These features lay the foundation for subsequent high-level feature extraction and semantic understanding in later layers. As these features relatively less complex and can be re-used across tasks, PEARL puts less emphasis on them through lower

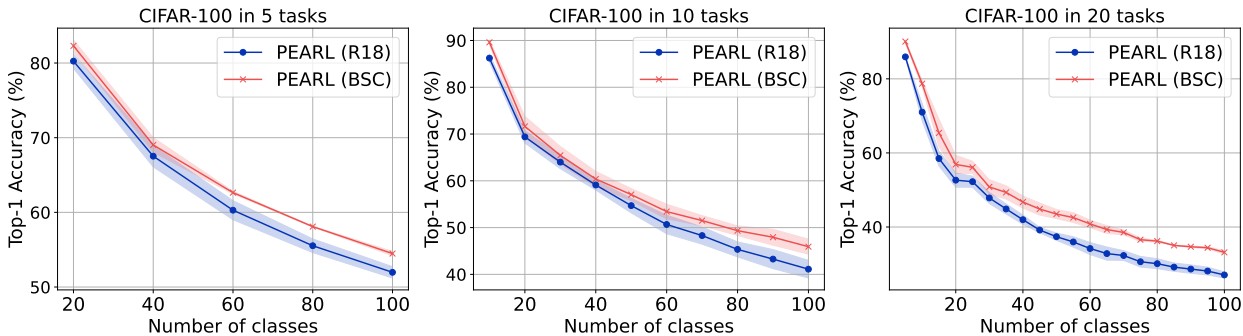

Figure 3: Evaluation of Class-IL ($A_T\%$) performance under longer task sequences in Seq-CIFAR100. In line with other parameter isolation approaches, there is a steady drop in performance as we move from 5 tasks to 10 tasks and eventually to 20 tasks.

rank allocation. On the other hand, intermediate layers handle more abstract and complex features, such as parts of objects and entire objects, necessitating higher rank allocation. Therefore, PEARL puts a major emphasis on intermediate layers with higher rank allocation. The top layers of the ViT model are designed to capture high-level semantic features and global contextual information. By later layers, the model has likely already captured the most significant semantic features relevant to the task, and additional layers contribute marginally smaller improvements to the final output quality. Therefore, PEARL allocates a lower threshold and a corresponding lower rank in later layers. In line with the aforementioned findings in Zeng et al. (2024), PEARL displays a bell-curve shaped dynamic rank allocation across layers thereby optimizing resource allocation and maximizing information reuse.

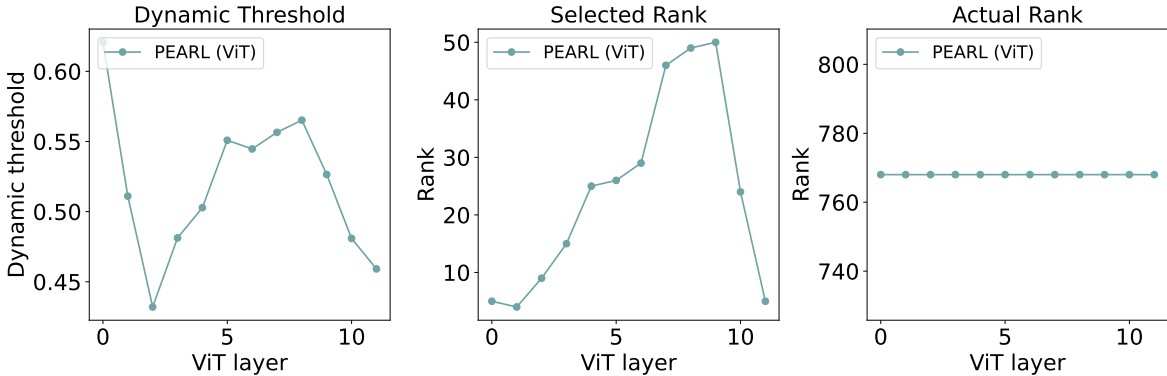

Figure 4: Depiction of final task dynamic resource allocation in ImageNet-R (5T). (left) Dynamic threshold of different ViT layers based on the similarity of current task with respect to reference task for deciding the rank. (middle) Selected rank based on dynamic threshold. (right) Actual rank of Key weight matrix. As can be seen both dynamic threshold and selected rank show more emphasis for later layers, in line with the actual rank.

### C.7 Additional comparison with adapter-based methods

Table 2 provides an extensive comparison of different adapter-based and LoRA-based PEFT CL approaches. In addition, Table 14 provides comparison against additional adapter-based methods in the Class-IL setting on ImageNet-R (10T) using a ViT-B/16 backbone pre-trained on ImageNet-21K. As can be seen, PEARL (ViT) produces superior performance across both the metrics and beats the baselines by a large margin. Although LoRA-rank allocation is dynamic in our framework, PEARL (ViT) leaves slightly more number of

parameters compared to adapter-based approaches. Although adapter-based methods are parameter efficient, We note that their performance is sub-par and can be even worse in longer task sequences.

Table 14: Following Tan et al. (2024), we compare additional adapter-based methods in the Class-IL setting on ImageNet-R (10T) using a ViT-B/16 backbone pre-trained on ImageNet-21K.

| Method | $A_T$ (%) ↑ | $\bar{A}$ (%) ↑ | Params (M) |
|---|---|---|---|
| Adam-ssf (Zhou et al., 2024a) | $66.61 \pm 0.09$ | $74.36 \pm 1.00$ | 0.20 |
| LAE (Gao et al., 2023a) | $72.29 \pm 0.14$ | $77.99 \pm 0.46$ | **0.19** |
| ADA (Ermis et al., 2022) | $73.76 \pm 0.27$ | $79.57 \pm 0.84$ | 1.19 |
| SSIAT (Tan et al., 2024) | $79.38 \pm 0.59$ | $83.63 \pm 0.43$ | 1.19 |
| PEARL | $\mathbf{88.76 \pm 0.69}$ | $\mathbf{91.67 \pm 0.69}$ | 1.75 |

## C.8  Additional comparison with prompt-based methods

Table 15 provides an extensive comparison of prompt-based methods in the continual learning (CL) setting on ImageNet-R. Following HiDe-Prompt (Wang et al., 2023a), we evaluate our method using the final forgetting measure (FFM) and cumulative forgetting measure (CFM), defined as follows. Let $A_{i,t}$ denote the accuracy on the $i$-th task after learning the $t$-th task.

After learning all $T$ tasks, the forgetting metrics are defined as:

$$FFM = \frac{1}{T-1} \sum_{i=1}^{T-1} \max_{t \in \{1,...,T-1\}} (A_{i,t} - A_{i,T}) \tag{A58}$$

$$CFM = \frac{1}{T-1} \frac{1}{T-1} \sum_{j=2}^{T} \sum_{i=1}^{j-1} \max_{t \in \{1,...,j-1\}} (A_{i,t} - A_{i,j}) \tag{A59}$$

As shown in the results, PEARL (ViT) consistently achieves the best overall performance across both FFM and CFM metrics. HiDe-Prompt and EvoPrompt rank second and third overall, respectively, across all metrics and task settings. Notably, our method demonstrates stable and competitive performance across varying task numbers (5, 10, and 20), highlighting its robustness in different continual learning scenarios.

Table 15: Following Wang et al. (2023a), we compare additional prompt-based methods in the Class-IL setting on ImageNet-R using a ViT-B/16 backbone pre-trained on ImageNet-21K. **Bold** indicates the best result and underline indicates the second-best result.

| Method | 5 Task @ 40 classes/task | | 10 Task @ 20 classes/task | | 20 Task @ 10 classes/task | |
|---|---|---|---|---|---|---|
| | FFM | CFM | FFM | CFM | FFM | CFM |
| L2P (Wang et al., 2022e) | $3.94 \pm 0.16$ | $3.55 \pm 0.20$ | $5.01 \pm 0.40$ | $4.41 \pm 0.43$ | $10.76 \pm 0.45$ | $7.88 \pm 0.17$ |
| DualPrompt (Wang et al., 2022d) | $3.32 \pm 0.16$ | $2.78 \pm 0.25$ | $4.61 \pm 0.07$ | $3.70 \pm 0.18$ | $7.30 \pm 0.18$ | $5.16 \pm 0.34$ |
| CODA-P (Smith et al., 2023b) | $8.89 \pm 0.65$ | $7.65 \pm 0.98$ | $7.94 \pm 0.08$ | $6.72 \pm 0.79$ | $8.23 \pm 0.86$ | $6.95 \pm 0.70$ |
| LGCL (Khan et al., 2023) | $3.04 \pm 0.36$ | $2.50 \pm 0.38$ | $4.75 \pm 0.33$ | $3.38 \pm 0.58$ | $7.35 \pm 0.65$ | $5.05 \pm 0.32$ |
| HiDe-Prompt (Wang et al., 2023a) | $3.15 \pm 0.46$ | $2.64 \pm 0.16$ | $\mathbf{2.29 \pm 0.27}$ | $2.33 \pm 0.17$ | - | $\mathbf{2.23 \pm 0.38}$ |
| PGP Qiao et al. (2024) | $3.36 \pm 0.23$ | $2.85 \pm 0.25$ | $4.53 \pm 0.40$ | $3.63 \pm 0.35$ | $7.17 \pm 0.21$ | $5.09 \pm 0.25$ |
| EvoPrompt (Kurniawan et al., 2024) | $\mathbf{1.79 \pm 0.31}$ | $1.41 \pm 0.32$ | $4.22 \pm 0.42$ | $3.59 \pm 0.52$ | $6.72 \pm 0.90$ | $5.67 \pm 0.26$ |
| CPrompt (Gao et al., 2024) | $6.00 \pm 0.00$ | $5.49 \pm 0.00$ | $6.10 \pm 0.75$ | $5.60 \pm 1.35$ | $5.98 \pm 0.24$ | $5.54 \pm 0.48$ |
| ConvPrompt (Roy et al., 2024) | $3.42 \pm 0.05$ | $2.36 \pm 0.16$ | $4.17 \pm 0.04$ | $3.11 \pm 0.17$ | $\mathbf{4.87 \pm 0.57}$ | $3.57 \pm 0.25$ |
| PEARL | $2.28 \pm 0.13$ | $\mathbf{0.78 \pm 0.07}$ | $3.63 \pm 0.17$ | $\mathbf{1.57 \pm 0.11}$ | $5.93 \pm 0.25$ | $2.49 \pm 0.07$ |

## C.9  Inference time comparison in Class-IL

Table 16 presents an overview of per-image inference times on ImageNet-R (20T) for various baselines and PEARL (ViT). It is evident that adapter-based methods exhibit slower growth compared to LoRA-based

methods, resulting in lower computational effort during inference. However, as shown in Table 2 and 14, adapter-based approaches experience a higher degree of forgetting, leading to reduced performance across tasks. In contrast, LoRA-based methods, including PEARL (ViT), involve a slightly larger number of parameters and necessitate forward passes through each sub-network during inference. While PEARL (ViT) achieves superior performance across tasks, the additional overhead during inference remains a limitation of LoRA-based approaches, including our own.

Table 16: Per-image inference time on ImageNet-R (20T) for each method.

| Method | Inference Time (ms) |
| --- | --- |
| L2P Wang et al. (2022e) | 9.53 |
| DualPrompt Wang et al. (2022c) | 9.51 |
| CODA-P Smith et al. (2023b) | 9.47 |
| Adam-ssf Zhou et al. (2024a) | 10.03 |
| SEMA Wang et al. (2025) | **7.39** |
| PEARL (ViT) | 14.67 |

### C.10 Task-wise resource allocation

As PEARL incorporates dynamic rank allocation based on task proximity in parameter space, we expect to see a non-uniform distribution of resources across different tasks in CL. Figure 5(a) presents an overview of distribution of normalized resource allocation across tasks for all PEARL variants. As can be seen, resource allocation is quite dynamic with highest variance seen in PEARL (ViT) in ImageNet-R (10T). Guided by the task proximity in the parameter space, PEARL dynamically allocates resources thereby ensuring optimum trade-off between performance and model growth. In addition, we also provide a comparison of normalized resource allocation across three runs of PEARL (ViT) on ImageNet-R (10T) in Figure 5(b). We duly note that PEARL relies solely on the proximity of a particular task with respect to the reference task weights in the parameter space. We also clarify that task proximity as defined in PEARL does not correlate with the semantic similarity of tasks. With different random seeds, the training trajectory and thus the proximity of tasks with respect to the reference task changes. Therefore, the distribution of parameter allocation exhibits significant variation across tasks in different runs.

## D  Experimental setup

### D.1  Backbones

To demonstrate the versatility of PEARL across different architectures, we also explore three distinct backbones: (i) Convolutional networks such as ResNet-18 and a ResNet-18-like architecture incorporating Blueprint Separable Convolutions (BSC) (Li et al., 2022b), and (ii) the Vision Transformer (ViT) Base 16 pre-trained on ImageNet-21k.

With regard to the PEARL (R18), the backbone is same as the one used in Mammoth CL benchmark (Buzzega et al., 2020). In line with previous works, the backbone is randomly initialized and trained for CL tasks. However, we introduce PyTorch ModuleLists all throughout to make it easier to add / remove convolutional and BatchNorm layers at will. The ResNet-18 backbone consists of 4 layers with 2 blocks each. Each block in turn consists of two 3x3 convolutional layers with a BatchNorm and ReLU activation in between and after. ResNet-18 also entails shortcut connections in the form of a 3x3 convolutional layer followed by a BatchNorm layer and ReLU activation. Within PEARL (R18), we reserve separate classification and BatchNorm layers for each task. As the backbone is not pre-trained, we treat the first task as the reference task and conduct dynamic allocation from the second task onward. LoRA components are allocated for every convolutional layer. Their rank is dynamically decided after SVD decomposition of respective task vectors. Even after dynamic allocation and further training, LoRA components, BatchNorm layers and a corresponding classification layer are kept separate for each task.

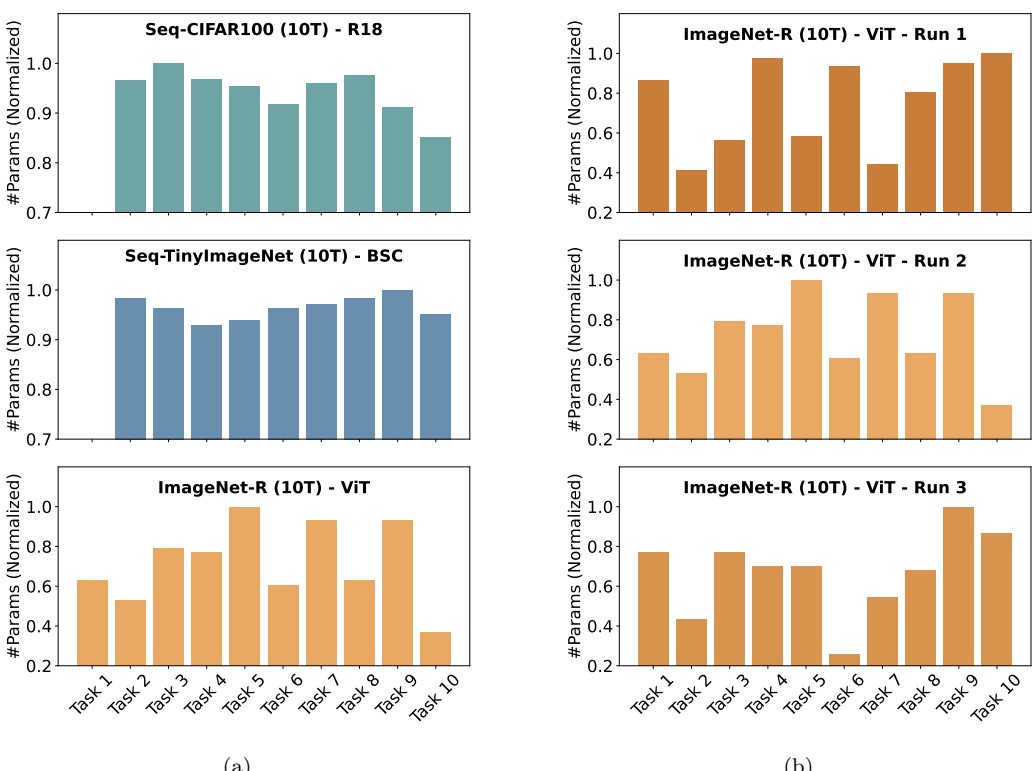

Figure 5: Task-wise resource allocation in parameter space (normalized number of parameters) for PEARL variants. (a) Visualizes resource allocation across Seq-CIFAR100 (10T), Seq-TinyImageNet (10T), and ImageNet-R (10T). (b) Shows resource allocation dynamics for ImageNet-R (10T) using ViT across three runs.

Li et al. (2022b) introduced blueprint separable convolutions (BSC) to optimize and reduce redundancy in convolutional operations. Essentially, BSC entails a 1x1 operation (point-wise filter) followed by a depth-wise convolution layer. This separability reduces the memory and computational footprint resulting far more efficient model. Such a model is well suited for parameter isolation CL approach as it reduces overall model size when dealing with longer task sequences. PEARL (BSC) backbone consists of 3 layers with 2 blocks each. Within each block, the organization of layers is same as that of ResNet-18 except that traditional convolutional layers are replaced by BSC layers. As the BSC backbone is a custom, small network, pre-trained model is not readily available. Therefore, we treat the first task as the reference task and conduct dynamic allocation from the second task onward. LoRA components are allocated for every point-wise filters as they have the majority of the footprint in terms of number of parameters. Since depth-wise filters have relatively lower footprint, we allocate LoRA components and select the rank of the task vector after decomposition. Same as in ResNet-18, even after dynamic allocation and further training, LoRA components, BatchNorm layers and a corresponding classification layer are kept separate for each task.

For PEARL (ViT), we use a ViT-B/16 backbone pre-trained on the ImageNet-21K dataset. To support our approach, PyTorch ModuleList structures are employed to facilitate the integration of task-specific LoRA components. LoRA weights are inserted in all layers of the ViT backbone, and as shown previously in Table 3, adding LoRA weights to the Key matrix yields better results. The rank of these LoRA components is dynamically determined by performing SVD decomposition on the corresponding task vectors. During inference, parameter-efficient task-specific LoRA weights that are used to predict image labels in both Task-IL and Class-IL scenarios.

### D.2 Datasets and settings

We evaluate PEARL across two distinct scenarios of CL: Class-IL and Task-IL. In both these settings, the model encounters a new set of classes in each task and must learn to distinguish all classes encountered thus far after each task. In practice, we split CIFAR-100 (Krizhevsky et al., 2009), TinyImageNet (Le & Yang, 2015), and ImageNet-R (Hendrycks et al., 2021) into different tasks depending on the number of tasks as in Table 17.

Table 17: Dataset organization across different tasks and settings.

| Dataset | Setting | Image size | Tasks | Classes per task |
|---|---|---|---|---|
| Seq-CIFAR100 | 5T | 32x32 | 5 | 20 |
| | 10T | 32x32 | 10 | 10 |
| | 20T | 32x32 | 20 | 5 |
| Seq-TinyImageNet | 5T | 64x64 | 5 | 40 |
| | 10T | 64x64 | 10 | 20 |
| ImageNet-R | 5T | 224x224 | 5 | 40 |
| | 10T | 224x224 | 10 | 20 |
| | 20T | 224x224 | 20 | 10 |

PEARL is a CL framework that addresses catastrophic forgetting by using parameter isolation through the dynamic allocation of LoRA components. It creates a distinct sub-network for each task. When a new task is introduced, a sub-network is instantiated from the reference task and trained using a cross-entropy loss function. After a few epochs of fine-tuning on the current task, task vectors are computed and decomposed using SV-Decomposition. The LoRA components are initialized with a rank that is determined dynamically according to the criteria set forth in our approach. These components are then re-initialized and trained for the current task. Once the training for a task is complete, the associated sub-network—which includes the LoRA components, BatchNorm layers, and classification layer—is frozen.

The model is trained simultaneously on both Class-IL and Task-IL settings, as their training regimes are identical. During inference in the Task-IL setting, the appropriate task-specific LoRA weights are selected based on the given task identity, and its output is inferred for maximum activation. However, inference in the Class-IL setting is more complex, as no task-identity is available. In this case, each test image passes through every sub-network, and their respective classifier outputs are concatenated. Since each task-specific sub-network is trained independently and the activation magnitudes produced can be imbalanced, the performance in the Class-IL setting often lags significantly behind that in the Task-IL setting. Although PEARL is highly efficient in terms of parameter allocation, it introduces inference overhead in Class-IL scenarios, similar to C-LoRA. Unlike InfLoRA, where the learned parameters can be folded back into the original model weights for more efficient inference, PEARL stores task-specific LoRA sub-networks for each task. This task-wise separation, while effective for reducing interference between tasks, results in slightly increased storage and retrieval costs during inference, especially as the number of tasks grow beyond a reasonable limit.

### D.3 Training details

For experiments concerning PEARL (ViT) on the ImageNet-R dataset, we use the Adam optimizer (Kingma & Ba, 2015) with hyperparameters $\beta_1 = 0.9$ and $\beta_2 = 0.999$. The batch size is set to 128. The ViT model is fine-tuned on the current task with a learning rate of $5 \times 10^{-3}$ for 25 epochs ($E_1$). Following this, the LoRA weights are trained separately for an additional 25 epochs ($E_2$) using a learning rate of $5 \times 10^{-4}$. We observed that a cosine-decaying learning rate schedule outperforms a constant learning rate. Regarding the LoRA parameterization, the rank is determined dynamically based on the criteria outlined in our approach, and the LoRA alpha parameter is set to twice the rank.

With regard to PEARL convolutional variants, we use a batch size of 32 for all experiments. Furthermore, we fine-tune the reference task weights for 15 epochs ($E_1$) with a best learning rate found in Section C.3. Following this, the LoRA components are initialized with dynamic ranks and are fine-tuned further for 50 epochs ($E_2$).

---

**Algorithm 2** PEARL (without pre-training)

---

1: **Input**: Randomly initialized model $\Phi_\theta$, Data distributions $\{\mathcal{D}_t\}_{t=1}^K$, Epochs $E_1$, $E_2$
2: Reference task training using Eqn. 1
3: $\theta_r = \{f_{\theta^r}, g_{\theta^r}\}$, $\theta = \theta_r$
4: **for** tasks $t \in \{2, .., K\}$ **do**
5:     $\theta_t \leftarrow \{\text{deepcopy}(f_{\theta^r}), g_{\theta^t}\}$
6:     $f_{\theta^t} \leftarrow \{\}$
7:     Fine-tune $\theta_t$ on $\mathcal{D}_t$ for $E_1$ epochs
8:     **for** target layers $l \in \{1, 2, .., L\}$ **do**
9:         Compute $\mathcal{W}_{cl}^t = \mathcal{W}_l^t - \mathcal{W}_l^r$
10:         Decompose $\mathcal{W}_c^t$ using Eqn. 2
11:         Find dynamic rank $k_l$ (Eqn. 3 & Eqn.4)
12:         Initialize LoRA components $B_l$ and $A_l$ (Eqn. 5)
13:         $f_{\theta^t} = f_{\theta^t} \cup \{B_l, A_l\}$
14:     Re-initialize task-specific weights $\{f_{\theta^t}, g_{\theta^t}\}$
15:     $\theta \leftarrow \theta \cup \{f_{\theta^t}, g_{\theta^t}\}$
16:     Fine-tune $\{f_{\theta^t}, g_{\theta^t}\}$ on $\mathcal{D}_t$ for $E_2$ epochs
17: **return** model $\Phi_\theta$

---

### D.4 PEARL Algorithm in the absence of pre-training

Algorithm 1 provides an account of PEARL's functioning when a pre-trained model is readily available. However, pre-training can be a luxury for custom, compact models. Therefore, we design PEARL to seamlessly work when pre-training is not available. Algorithm 2 illustrates the functioning of PEARL in the absence of pre-training. The only difference being, the first task is treated as a reference task and all the subsequent tasks leverage first task information for efficient CL. Although it is possible to use more number of classes in the first task for better knowledge consolidation, first task in PEARL contains same number of classes as the rest of the tasks.

