# OpenReview forum: "Parameter Efficient Continual Learning with Dynamic Low- Rank Adaptation"
_TMLR — Accepted by TMLR_

### Review · Reviewer_teHE · 2025-11-07

**Summary Of Contributions:**

This paper proposes Parameter-Efficient Adaptive Ranking for lifelong learning with LoRA (PEARL), a rehearsal‑free continual learning (CL) framework that dynamically allocates the LoRA rank per task and per layer using information from reference task weights. For each new task, the method first fine‑tunes a copy of the reference network to obtain (current) weights. Different from standard LoRA, the method computes a dynamic threshold to reflect proximity between the current and reference weights, and then accordingly sets a layer‑wise rank to fine-tune the parameters through LoRA to avoid catastrophic forgetting while keeping learning new tasks.

Strengths/contributions

The paper is well written. The proposed data-driven rank selection method for LoRA is simple to apply and works with or without pre-training. It also shows strong empirical results across models and tasks. Besides the performance, the paper further provides analyses on several aspects (e.g., static vs. dynamic ranks, forgetting metrics and re-initialization benefits).

Weaknesses

(i) If I understand correctly, the proximity score $T$ is used as a threshold in [0,1] for Eq.(4), but the given normalization may exceed 1. This lacks more explanations or clarification.

(ii) My major concern is about the computation (also mentioned in Section 5): the method requires a full‑model fine‑tune per task before training LoRA, I was wondering if in practice how much additional computation cost is introduced compared to a standard LoRA (with fixed rank)?

(iii) As mentioned in Section 5, the model assumes knowledge of task boundaries, which restricts the practical applicability of it.

**Audience:**

Yes

**Audience Explanation:**

Dynamic, proximity-aware LoRA rank allocation is timely for the CL and PEFT communities. The method is evaluated on CNNs and ViTs, spans both Class-IL and Task-IL settings, and delivers practical procedures with sizable gains. Comprehensive ablations and analyses further inform future research and applications.

**Broader Impact Concerns:**

No concern.

**Claims And Evidence:**

Yes

**Claims Explanation:**

This paper presents comprehensive performance experiments, together with analyses and ablation studies that provide additional insights (see Section 4 and Appendix B).

**Requested Changes:**

(i) Clarify the proximity score $T$ as discussed in the weaknesses.
(ii) Discuss the computational cost of Algorithm 1. (Experiments are not necessary)

---

> ### Author Response · Authors · 2025-12-02
> **Reply to Reviewer teHE**
>
> We sincerely appreciate the reviewer’s thoughtful feedback and recognition of the strengths of our paper. Please find our responses to raised concerns below:
>
>
> > If I understand correctly, the proximity score is used as a threshold in [0,1] for Eq.(4), but the given normalization may exceed 1. This lacks more explanations or clarification.
>
> We regret the lack of clarity surrounding the dynamic threshold criterion defined in Equation. 3. Indeed, T is not guaranteed to lie in [0,1] and T can be as large as 2 in extreme cases. We regret that we inadvertently omitted this detail in our manuscript. In our implementation, whenever T>1, we simply clip it to 1, ensuring that the cumulative-variance condition in Eq. 4 is always well-defined. We will clarify this behavior explicitly in the final revision and update the text to note both the possible range of T and the clipping step.
>
> Figure 2 depicts the dynamic thresholds of different convolutional layers for the final task in Seq-CIFAR100 (5T). As can be seen, the dynamic threshold threshold criterion within PEARL is quite conservative in its estimates. Moreover, PEARL achieves a best trade-off between model size and performance without extensive tuning of rank (see Table 4.) Although theoretically T can be as large as 2 and is clipped when T>1, we did not encounter such a high threshold in any of our experiments.
> > My major concern is about the computation (also mentioned in Section 5): the method requires a full‑model fine‑tune per task before training LoRA, I was wondering if in practice how much additional computation cost is introduced compared to a standard LoRA (with fixed rank)?
>
> PEARL leverages the information contained in the reference task weights and dynamically determines the rank of task-specific, layer-specific LoRA components during training based on the current task’s proximity to reference task weights in parameter space. To achieve this, PEARL requires full-model fine-tuning to estimate task vectors and thereby compute the dynamic rank. In our setup, the number of epochs used for full-model fine-tuning is a configurable hyperparameter, and we fix it to 15 epochs for ImageNet-R. Beyond this step, the remaining training procedure is identical to standard LoRA (with a fixed rank), including the number of epochs used to fine-tune the LoRA components.
>
> Relative to standard fixed-rank LoRA, PEARL incurs a 1.6x increase in training time, regardless of the number of tasks. We agree with the reviewer that performing full-model fine-tuning for each task prior to LoRA fine-tuning is a limitation of PEARL, and we explicitly acknowledge this in the Limitations section. As highlighted in the Conclusion, extending our approach to more realistic continual learning scenarios where task boundaries are unknown, and developing alternatives for efficiently estimating task vectors without fully fine-tuning the pretrained model, constitute promising directions for future research.
>
>
> > As mentioned in Section 5, the model assumes knowledge of task boundaries, which restricts the practical applicability of it.
>
> We agree with the reviewer and acknowledge reliance on task boundaries as a limitation and admit in our manuscript that extending our work to more realistic settings such as general CL scenarios where task boundaries are not known at training time is one of the useful follow up research directions for this work. However, we also point out that the majority of the baselines considered in this work rely extensively on the task boundary. Therefore, reliance on task boundaries is a general limitation of existing CL works, not just limited to ours.
>
> Although not explored, PEARL can be easily adapted to work for online CL settings by leveraging the training dynamics of loss values to enable the automatic recognition of the data distribution shifts, akin to [1]. As long as we have a notion of task separation, either given or estimated, PEARL works exactly the same without requiring any changes in its architecture or training paradigm.
>
> In addition, as suggested by the reviewer, we will include clarification on  the proximity score and discuss the computational cost of PEARL compared to standard LoRA (with a fixed rank) in the final revision.
>
> We sincerely appreciate the reviewer’s valuable feedback and are eager to incorporate their recommendations. We believe our clarifications have strengthened your confidence in our work. If any concerns remain, please let us know for further clarification.
>
>
> [1] Wei, Xiwen, Guihong Li, and Radu Marculescu. "Online-lora: Task-free online continual learning via low rank adaptation." 2025 IEEE/CVF Winter Conference on Applications of Computer Vision (WACV). IEEE, 2025.

---

> > ### Comment · Reviewer_teHE · 2025-12-11
> >
> > I appreciate the authors’ detailed responses to my comments and have also considered their replies to the other reviewers. I recommend acceptance for publication.

---

### Review · Reviewer_22Px · 2025-11-16

**Summary Of Contributions:**

The authors propose a rehearsal free CL method (PEARL) that uses finetuning with LoRA modules and assigns a different low-rank to each task and layer. The main idea is to treat a pre-trained model (or the first CL task) as a “reference network” and for each subsequent task to compute a task vector (diff between current fine-tuned weights and reference weights). An SVD of this task vector gives singular values whose cumulative energy is compared against a normalized distance between current task and reference. This yields a layer-specific dynamic rank, aiming to increase plasticity for layers where the new task induces large weight changes, and reduce plasticity where the task is closer to the reference. These LoRA modules are then re-initialized and trained only for the current task, while the reference network remains fixed.

**Audience:**

Yes

**Audience Explanation:**

CL is an active research area and the sub-area of replay-free parameter-efficient methods is of emerging interest. So yes, I have no doubt that the paper will be found interesting by some researchers.

**Broader Impact Concerns:**

I do not see any such concerns.

**Claims And Evidence:**

Yes

**Claims Explanation:**

The main empirical claims are supported by the reported experiments. The implementation appears technically sound at a high level. There are some gaps however in the definition of the dynamic threshold. See below:

a) the fraction in Eq4 (where the denominator is the total singular-value mass) is in [0,1] but T is not guaranteed to lie in [0,1]. T can be as large as 2 in extreme cases. The paper does not mention clipping T to [0,1] or handling the case T>1 and Eq4 is ill-defined if the inequality “cumulative variance ≥ T” never holds.

b) Also the +1 in Eq4 is not motivated. It forces the chosen rank k to overshoot the minimum rank satisfying the variance criterion, which may or may not be desirable. Plz explain.

**Requested Changes:**

- plz clarify and correct the definition of the threshold T. The paper should state T's range, explain the behavior when T>1, and specify whether T is clipped or normalized.

- Provide a justification for the “+1” term in the rank selection rule. the motivation should be explained or the formula revised.

- *Important* Include a comparison against at least one recent dynamic-rank LoRA CL method (e.g., FM-LoRA, CoDyRA). Both methods are highly relevant and address very related problems.

- Explicitly analyze (or at least discuss) the potential sensitivity of the dynamic rank to noise in the SVD of Wc, especially for small matrices or early convolutional layers where task vectors may be low-energy or poorly conditioned.

- Provide an ablation isolating the contribution of the “reference task” choice (e.g., first task vs pre-trained model). It is VERY important to see whether the dynamic ranks or performance depend strongly on which task is used as reference.

- Many baseline results are taken from prior papers with different training pipelines. At minimum, report the exact data preprocessing and augmentation used for PEARL and explain any known differences from the baselines’ setups.

- Class-IL inference requires passing each test sample through all task-specific branches. Inference time grows linearly with the number of tasks. It would be useful to quantify this cost and compare it to alternative PEFT-CL approaches that avoid multi-branch inference.

- Provide a clearer comparison of parameter counts versus rehearsal methods. Although rehearsal methods use a buffer, PEARL uses per-task adapters. It would be helpful to include a table converting both into a unified memory footprint metric.

- Expand the analysis of failure modes. For example are there cases where PEARL selects very small ranks and significantly underfits or very large ranks that approximate full fine-tuning? A controlled experiment showing such cases would help show the limitations.

- Improve the clarity around re-initialization. Currently LoRA components are initialized from the SVD and then immediately re-initialized. The manuscript should explain when and why the SVD-based initialization is beneficial given that it is then discarded.

---

> ### Author Response · Authors · 2025-12-02
> **Reply to Reviewer 22Px (1/3)**
>
> We greatly appreciate the reviewer's considerate feedback and acknowledgment of the strengths of our paper. Below are our responses to the concerns raised:
>
> > Clarify and correct the definition of the threshold T.
>
> We regret the lack of clarity surrounding the dynamic threshold criterion defined in Equation. 3. Indeed, T is not guaranteed to lie in [0,1] and T can be as large as 2 in extreme cases. We regret that we inadvertently omitted this detail in our manuscript. In our implementation, whenever T>1, we simply clip it to 1, ensuring that the cumulative-variance condition in Eq. 4 is always well-defined. We will clarify this behavior explicitly in the final revision and update the text to note both the possible range of T and the clipping step.
>
> Figure 2 depicts the dynamic thresholds of different convolutional layers for the final task in Seq-CIFAR100 (5T). As can be seen, the dynamic threshold threshold criterion within PEARL is quite conservative in its estimates. Moreover, PEARL achieves a best trade-off between model size and performance without extensive tuning of rank (see Table 4.) Although theoretically T can be as large as 2 and is clipped when T>1, we did not encounter such a high threshold in any of our experiments.
>
> > Also the +1 in Eq4 is not motivated.
>
> We thank the reviewer for highlighting this point. The “+1” term is indeed intentional. We add +1 to ensure that the chosen rank is always at least one singular value, preventing a degenerate zero-rank projection and preserving minimal expressive capacity. It serves as a safeguard, ensuring that each layer for every task has at least a minimal capacity to adapt features to the task distribution. Alternative strategies could achieve the same effect, and in practice, the term can be omitted if one wishes to be conservative about model growth. We will clarify the rationale in the final revision and discuss its effect on the selected rank.
>
> We agree with the reviewer that “+1” forces the chosen rank ‘k’ to overshoot the minimum rank satisfying the variance criterion. However, even with the “+1” rank, PEARL still achieves a best trade-off between model size and performance without extensive tuning of rank (see Table 4). Therefore, we argue that “+1” is necessary for training stability while having minimal impact on the model size. As per reviewer’s suggestion, we will clarify the same in the final revision.
>
> > Important Include a comparison against at least one recent dynamic-rank LoRA CL method
>
> Within Section A.3, we provide extensive discussion on PEFT LoRA counterparts, which includes PEFT approaches with both static as well as dynamic rank allocation. Overall, the approaches within dynamic rank allocation are not specifically tailored for CL and lack results for direct comparisons with PEARL variants. Although Dynamic Rank-Selective LoRA focuses on CL, it is tailored for vision-language models, limiting the possibility of a direct comparison with PEARL. To the best of our knowledge, PEARL is one of the early approaches tailored specifically for efficient custom, compact CL. We will try our best to include a direct comparison with at least one of the dynamic rank allocation approaches in the final revision.
>
> > Explicitly analyze (or at least discuss) the potential sensitivity of the dynamic rank to noise in the SVD of Wc
>
> We agree with the reviewer that PEARL is quite sensitive to the stability of task vector and noise contained in the resultant SVD decomposition thereof. In cases where task vectors are poorly conditioned, the dynamic rank estimation can be imprecise resulting in over or under allocation of dynamic rank. Figure 5(b) provides a depiction of  resource allocation dynamics for ImageNet-R (10T) using PEARL (ViT) across three distinct runs. We note that PEARL relies solely on the proximity of a task relative to the reference task weights in parameter space. It is important to clarify that task proximity, as defined in PEARL, does not correspond to semantic similarity between tasks. Different random seeds can alter the training trajectory, thereby changing the proximity of tasks relative to the reference task. As a result, the distribution of parameter allocation can vary significantly across tasks in different runs. Following the reviewer’s suggestion, we will provide additional clarification on the potential sensitivity of dynamic rank in PEARL in the final revision.

---

> > ### Author Response · Authors · 2025-12-02
> > **Reply to Reviewer 22Px (2/3)**
> >
> > > Provide an ablation isolating the contribution of the “reference task” choice.
> >
> > Table 6 provides a Class-IL performance comparison of PEARL (R18) with and without pre-training. In the former case, PEARL (R18) is pre-trained on Imagenet-1K, while in the latter case, PEARL (R18) is randomly initialized. Thanks to the versatility of PEARL, the framework can work with or without pre-training. As shown, pre-training on ImageNet-1K consistently improves performance across tasks and datasets. Whenever feasible, we recommend using a pre-trained backbone with PEARL to enhance generalization across tasks.
> >
> > > Many baseline results are taken from prior papers with different training pipelines.
> >
> >
> > We conduct extensive experiments across 3 different architectures (ResNet, Separable Convolutional Network, and Vision Transformer), 8 distinct settings, 2 CL scenarios, and 2 CL metrics to clearly demonstrate that PEARL significantly outperforms the considered baselines. To further substantiate our claims, we provide 10+ comprehensive analysis in Appendix B. As navigating faithful reproduction of results across a multitude of baselines is not easy, it is only natural that we re-purpose the results reported in the original papers to save our precious compute and efforts. To the best of our knowledge, we haven’t significantly swayed away from the CL settings, data preparation and metrics. We will update the final revision with footnotes in cases where experimental designs differ significantly with the baselines.
> >
> > > Class-IL inference requires passing each test sample through all task-specific branches.
> >
> > Table 16 presents an overview of per-image inference times on ImageNet-R (20T) for various baselines and PEARL (ViT). Note that this is an extreme case in our experiments where 20-tasks are involved. Adapter-based methods grow more slowly than LoRA-based methods, reducing inference cost but with higher forgetting (Tables 2 and 14). In contrast, LoRA-based methods, including PEARL (ViT), have slightly more parameters and require forward passes through each sub-network during inference. While PEARL (ViT) delivers superior performance across tasks, this additional inference overhead highlights a key limitation of LoRA-based approaches, including our implementation.
> >
> > > Provide a clearer comparison of parameter counts versus rehearsal methods.
> >
> > PEARL is aimed at addressing the problem of optimum rank selection in LoRA-based PEFT continual learning approaches. Table 4: Class-IL performance comparison of PEARL with static rank allocation for LoRA components. Within the same table, we also provide model size in terms of number of parameters. PEARL achieves a best trade-off between model size and performance without extensive tuning of rank. We also provide model growth in terms of the number of parameters across different numbers of tasks in Section 4.9. These analyses provide a clear view of how PEARL grows in size with number of tasks and rank. On the other hand, rehearsal methods scale on memory buffer size  and PEARL on parameter count, a direct comparison is nontrivial and beyond the scope of this study.
> >
> >
> >
> > > Expand the analysis of failure modes.
> >
> > Table 4 provides a Class-IL performance comparison of PEARL with static rank allocation for LoRA components. Increasing the static rank tends to boost performance but also increases the model size, and after a certain point the extra capacity yields diminishing returns. Therefore, selecting an appropriate rank is key to maintaining a favorable balance between model size and accuracy. PEARL addresses this by assigning ranks dynamically on a per-task and per-layer basis, allowing it to achieve strong performance while keeping model growth under control across different datasets and architectures.
> > Throughout our multitude of experiments across multiple architectures, CL datasets and scenarios, PEARL has shown a tendency to  achieve a best trade-off between model size and performance without extensive tuning of rank as in static rank allocation. Unfortunately, we haven’t observed in our experiments any failure modes where PEARL selects very low or very high rank comparable with finetuning.
> >
> >
> > > Improve the clarity around re-initialization
> >
> > Within section 4.5 we detail out the effect of weight re-initialization of LoRA components, BN layers (if any), and classification layers following SV-decomposition in PEARL. The findings show that re-initializing the weights yields slightly improved performance for ResNet-18 on Seq-CIFAR100 (5T), significant gains for ViT on ImageNet-R (10T), and an impressive two-fold boost for BSC on Seq-TinyImageNet (10T). Taken together, these results suggest that weight re-initialization benefits PEARL across a range of continual learning architectures and configurations over SVD-based initialization. The findings are in line with the literature (e.g. [1]) which emphasize that re-initialization improves plasticity during continual learning.

---

> > > ### Author Response · Authors · 2025-12-02
> > > **Reply to Reviewer 22Px (3/3)**
> > >
> > > We sincerely thank the reviewer for their insightful feedback and are enthusiastic about integrating their recommendations. We believe our clarifications have improved the overall confidence in our work. If there are any lingering concerns, please let us know for further clarification.
> > >
> > > [1] Dohare, Shibhansh, et al. "Loss of plasticity in deep continual learning." Nature 632.8026 (2024): 768-774.

---

> > > > ### Comment · Reviewer_22Px · 2025-12-08
> > > >
> > > > I appreciate the extensive responses that the authors have written to my comments (I also read the other reviews and the responses to those comments). I think that the rebuttal period has improved the paper and I propose that the paper is accepted for publication.

---

### Review · Reviewer_7EMy · 2025-11-27

**Summary Of Contributions:**

This paper targets the setup of continual learning (CL), primarily in a low-rank space where models struggle to retain previously learned knowledge while learning new tasks. The authors propose PEARL (expaded as Parameter-Efficient and Adaptive Resource Allocation for Continual Learning), that uses low-rank adaptations to modify pre-trained models in a resource-efficient way, reducing the disruption to the model's performance on previous tasks while adapting to new ones in a continual fashion.

Contributions:
* The paper introduces a new formulation for a dynamic rank allocation mechanism for LoRA components, which adapts the rank based on the similarity between the current and previous tasks. The primary motivation is to efficiently reuse information from earlier tasks by assigning lower ranks to earlier layers and higher ranks to task-specific later layers, which is claimed to ensure minimal computational load while maintaining task-specific adaptations as the learning continues for tasks in sequential order.
* Another key contribution is the task-specific LoRA components, which are applied exclusively to the key modules of transformer models (though they have been explored in retrieval, knowledge merging, etc. in the past), the presented formulation focuses on being efficient for long task sequences in continual settings.
* The proposed method, PEARL, also leverages SV-decomposition of task vectors to initialize the LoRA components, for a strong starting point in task adaptation. This initialization is claimed to accelerate convergence.

Strengths

* One of the proposed methods primary focus is on resource efficiency. The dynamic rank allocation and task-specific LoRA components allow it to adapt to new tasks without a significant increase in model size. The detailed set of experiments are provided that demonstrates robust performance across multiple architectures (e.g., ResNet-18 and Vision Transformers) and tasks, outperforming other parameter-isolation techniques in terms of both accuracy and efficiency (as shown in the tables).

* Another notable strength is that the proposed formulation helps in the ability to benefit from pre-training, which adds to the model’s generalization ability and helps it perform well across multiple tasks, and may be helpful for the CL community.

* The limitations highlighted in the paper capture the open ends both in terms of application and experimentation.

Weaknesses:

* There are some lines of work that also explore LoRA on similar lines (e.g., Chaudhry et al., 2020; Wang et al., 2023; Farajtabar et al., 2020), making use of orthogonal subspaces formed using SVD formulation. The current version of the paper does not discuss the initial works/prior arts/parallel works in detail, and it would be good to add a comparison (empirically, if possible) or a discussion on similar lines to help the reader understand the primary gap that the proposed method fills.

* In the presented method, the task vector initialization assumes a precise measure of task similarity, but in practice, there can be multiple similarity criteria that are left underexplored, moreover, the accurate task vector estimation is a challenging problem, and more evaluation on that front would be good to add in the paper for more reliability in the presented approach.


[1] Wang, Xiao, et al. "Orthogonal subspace learning for language model continual learning." arXiv preprint arXiv:2310.14152 (2023).

[2] Chaudhry, Arslan, et al. "Continual learning in low-rank orthogonal subspaces." Advances in Neural Information Processing Systems 33 (2020): 9900-9911.

[3] Farajtabar, Mehrdad, et al. "Orthogonal gradient descent for continual learning." International Conference on Artificial Intelligence and Statistics. PMLR, 2020.

**Audience:**

Yes

**Audience Explanation:**

* The paper presents a new framework that is rehearsal-free, efficient, and adaptable across architectures.
* The paper show empirical results highlighting gains over the compared state-of-the-art methods.
* In terms of audience, the paper also has a Broad applicability in current literature, since handling multiple sequential tasks without catastrophic forgetting is essential for scaling ML systems in the real world.

In general, the paper targets a relevant problem in continual learning, and the increasing focus on efficient adaptation in large pre-trained models being the primary focus aligns with the TMLR scope.

**Broader Impact Concerns:**

The paper does not contain any explicit Broader Impact Statement section. However, given the scope of work, there are no major concerns about the ethical implications of the work that would require adding a Broader Impact Statement.

**Claims And Evidence:**

No

**Claims Explanation:**

* The results in Tables 1 and 2 show that the presented method gives performance improvements when compared to existing approaches across different scenarios (Class-IL, Task-IL, and with different architectures like ResNet-18, ViT, and BSC).
* The paper also includes suitable ablation studies (e.g., Table 3 and Figure 2) that show the importance of dynamic rank allocation for LoRA components, illustrating how it optimally balances model size and performance.

Overall, the experimental setup, including detailed comparison across multiple datasets and tasks, with ablation experiments justify the primary motivation/claims made in the paper.

**Requested Changes:**

Minor Changes/Edits:

* The algorithms, both 1 and 2, could be explained in more detail; in the current version, it becomes difficult to get the entire thing in one go. Several steps (e.g., the exact data flow between reference-task parameters and newly allocated LoRA modules, the re-initialization step, and the treatment of normalization layers) are a little ambiguous. It would be good to add precise definitions, and clarify the sequence of operations would make the methodology easier to reproduce.

* In the finetuning schedules, the difference between E1 and E2 is not completely clear; it would be good to describe whether reference-task weights are frozen or partially updated during different stages. Since in training/finetuning, there is a huge instability, and PEFT methods in general are a little sensitive to finetuning setups, it would be good to add all the details in the next iteration of the paper.

* The current description suggests passing every test sample through all task-specific sub-networks. This has an added inference cost, especially for large ViT models. If possible, it would be good to explicitly quantify inference-time FLOPs and latency, comparing them with baselines for a more clear picture of the improvements, which would also help in the utility of this work for future works.

---

> ### Author Response · Authors · 2025-12-02
> **Reply to Reviewer 7EMy**
>
> We sincerely thank the reviewer for their thoughtful feedback and for recognizing the strengths of our paper. Below are our responses to the concerns raised:
>
> > Discuss the initial works/prior arts/parallel works in detail on orthogonal subspaces
>
> Thank you for pointing out these relevant parallel lines of work. The orthogonal-subspace formulations in Chaudhry et al. (2020), Farajtabar et al. (2020), and Wang et al. (2023) are indeed very interesting, and we appreciate the suggestion to connect them more explicitly to our approach. These works primarily leverage SVD-derived subspaces to maintain task separation, which complements our perspective of using task-vector–based proximity for capacity allocation. Highlighting these connections will help readers better understand how PEARL differs both conceptually and operationally—particularly in how it adapts rank dynamically rather than fixing or orthogonalizing subspaces. We will include a dedicated discussion of these works in the final revision, and where feasible, add a brief empirical comparison or qualitative analysis to situate PEARL more clearly within this family of methods.
>
> > The task vector initialization assumes a precise measure of task similarity, but in practice, there can be multiple similarity criteria that are left underexplored,
>
> We appreciate this thoughtful observation regarding task-vector initialization and similarity criteria. Exploring alternative formulations of task similarity is indeed an interesting direction, and we agree that different metrics may offer complementary insights. However, a thorough evaluation of multiple similarity functions would require substantial additional experimentation to reliably assess their utility across architectures, datasets, and CL scenarios. Given the scope of this work and the already extensive empirical study, we consider such an investigation beyond the present version. That said, we will add a discussion in the final revision acknowledging this limitation and outlining the exploration of richer similarity metrics as an important avenue for future work.
>
> > Algorithms, both 1 and 2, could be explained in more detail
>
> We regret the lack of clarity in the current presentation of Algorithms 1 and 2. We agree that providing more precise definitions and a clearer step-by-step sequence will greatly improve reproducibility. We will revise these algorithmic descriptions in the final version to include explicit notation, clarified data flow, and a more detailed explanation of each operation.
>
> > In the finetuning schedules, the difference between E1 and E2 is not completely clear; it would be good to describe whether reference-task weights are frozen or partially updated during different stages
>
> We regret the lack of clarity regarding the distinction between the E1 and E2 fine-tuning schedules. We will update the final revision to clearly describe the behavior of reference-task parameters across both stages, provide complete training details, and clarify the stability considerations associated with the schedules.
>
> > Inference cost: quantify inference-time FLOPs and latency, comparing them with baselines
>
> In Section B.9, we report per-image inference times on ImageNet-R (20T) for various baselines and PEARL (ViT). Overall, adapter-based methods demonstrate lower inference times compared to LoRA-based approaches. However, this comes at the cost of increased forgetting. While PEARL achieves superior performance across tasks, the additional computational overhead reflects a limitation common to LoRA-based methods, including our own.
>
>
> We sincerely thank the reviewer for their insightful feedback and are enthusiastic about integrating their recommendations. We hope that our clarifications have improved the overall confidence in our work. If there are any lingering concerns, please let us know for further clarification.

---

### Author Response · Authors · 2026-01-14
**Follow-up and Next Steps for TMLR Submission 5965**

We would like to kindly follow up on the status of our TMLR submission (Submission Number: 5965). Two reviewers have provided positive feedback and have recommended the paper for acceptance, while we have not yet received comments from the third reviewer or further updates regarding the final decision.

We fully understand the time and effort involved in the review process and greatly appreciate your work. We wanted to ask if there are any updates on the decision, and whether there are any next steps or additional actions required from our side at this stage.

Thank you very much for your time and consideration.

---

### Decision · Action_Editor_FdPW · 2026-01-14

**Recommendation:** Accept with minor revision

**Additional Comments:**

Please revise the final version to address the remaining points (e.g., clarified algorithmic descriptions) as promised in the rebuttal.

**Audience:**

Yes

**Audience Explanation:**

CL and PEFT based approaches to CL are active areas of research, and the proposed dynamic LoRA rank allocation directly targets a practical limitation of existing PEFT-based CL methods. Multiple reviewers agree that the approach is timely, well executed, and relevant to researchers working on continual learning, efficient fine-tuning, and large pretrained models.

**Claims And Evidence:**

Yes

**Claims Explanation:**

The reviewers have unanimous agreement that the empirical evidence is strong and appropriately supports the paper's claims. The method is evaluated across multiple architectures, datasets, and continual learning scenarios, with consistent gains over relevant baselines. The key technical concerns (e.g., rank-selection criterion, threshold definition, computational cost) raised in the initial reviewers were clearly addressed in the rebuttal and will be clarified in the final version. No reviewer maintains any serious unresolved doubts about correctness or validity.